

# Abiotic versus biotic controls on soil nitrogen cycling in drylands along a 3200 km transect

Dongwei Liu[1]*, Weixing Zhu[1, 2], Xiaobo Wang[1]*, Yuepeng Pan[3], Chao Wang[1], Dan Xi[1, 5], Edith Bai[1], Yuesi Wang[3], Xingguo Han[1], Yunting Fang[1, 4]

[1] Key Laboratory of Forest Ecology and Management, Institute of Applied Ecology, Chinese Academy of Sciences, Shenyang, 110016, China

[2] Department of Biological Sciences, Binghamton University, State University of New York, Binghamton, NY 13902

[3] State Key Laboratory of Atmospheric Boundary Layer Physics and Atmospheric Chemistry (LAPC), Institute of Atmospheric Physics, Chinese Academy of Sciences, Beijing, 100029, China

[4] Qingyuan Forest CERN, Chinese Academy of Sciences, Shenyang 110016, China

[5] College of Resources and Environment, University of Chinese Academy of Sciences, Beijing, 100049, China

*These authors contributed equally to this work.

**Corresponding Author:**

Weixing Zhu

Department of Biological Sciences, Binghamton University – State University of New York, Binghamton, NY 13902-6000

Phone: (607) 777-3218

Fax: (607)-777-6521

Email: wxzhu@binghamton.edu

Yunting Fang

Institute of Applied Ecology, the Chinese Academy of Science, No.72, Wenhua Road, Shenyang, P. R. China, 110016

Phone: +86-24-83970541

Fax: +86-24-83970300

Email: fangyt@iae.ac.cn



## Abstract

Nitrogen (N) cycling of drylands under changing climate is not well understood. Our understanding about N cycling over larger scales to date relies heavily on the measurement of bulk soil N, and the information about soil internal N transformations remains limited. The $^{15}N$ natural abundance ($\delta^{15}N$) of ammonium and nitrate can serve as a proxy record for the N processes in soils. To better understand the patterns and mechanisms of water availability on soil N cycling in drylands, we collected soils along a 3200 km dryland transect at about 100 km intervals in northern China, with mean annual

precipitation (MAP) from 36 mm to 436 mm. We analysed N pools and $\delta^{15}N$ of ammonium, dual isotopes ($^{15}N$ and $^{18}O$) of nitrate, and the microbial gene abundance associated with soil N transformations. We found that the N status and their driven factors were different on the two sides of MAP = 100 mm. In the arid zone with MAP below 100 mm, soil inorganic N accumulated, with a large fraction being of atmospheric origin. Ammonia volatilization was strong because of the higher soil pH. The abundance of microbial genes associated with soil N transformations was also significantly low. In the semiarid

zone with MAP above 100 mm, soil inorganic N concentrations were low and controlled mainly by biological processes, e.g., plant uptake and denitrification. The preference of soil ammonium to nitrate by the dominant plant species may enhance the possibility of soil nitrate loss *via* denitrification. Overall, our study suggest that the shifting from abiotic to biotic controls on soil N biogeochemistry under global climate changes would greatly affect N losses, soil N availability, and other N transformation processes in these drylands in China.


**Key words:** soil inorganic N; $^{15}N$ natural abundance; soil microorganisms; functional genes; spatial patterns

## 1 Introduction

Drylands cover approximately 41 % of the Earth's land surface and play an essential role in providing ecosystem service and

regulating carbon (C) and nitrogen (N) cycling (Hartley et al., 2007; Poulter et al., 2014; Reynolds et al., 2007). After water, N availability is the most important limiting factors to plant productivity and microbial processes in dryland ecosystems (Collins et al., 2008; Hooper and Johnson, 1999). Despite low soil N mineralization rates and N availability, soil N losses are postulated to be higher relative to N pools in dryland ecosystems compared to mesic ecosystems (Austin, 2011; Austin et al., 2004; Dijkstra et al., 2012). However, we still lack a fully understanding for the constraints of N losses in drylands, because

multiple processes contribute to N losses and the response of those processes to changing climate is highly variable (Nielsen and Ball, 2015). The precipitation regimes in drylands were predicted to change during the 21st Century (IPCC, 2013), and more extreme climatic regimes will make dryland ecosystems more vulnerable to enhanced drought in some regions and intensive rain in others (Huntington, 2006; Knapp et al., 2008). Therefore, improving our understanding on N cycling and their controls would greatly enhance our ability of predicting the responses of dryland ecosystems to global changes.





The $^{15}N$ natural abundance (expressed as $\delta^{15}N$) can provide critical information on the N cycle and thus assist in understanding ecosystem N dynamics over the large scales (Amundson et al., 2003; Austin and Vitousek, 1998; Houlton et al., 2006). The general pattern that the foliar and soil $\delta^{15}N$ increases as precipitation decreases has been found at both regional (Aranibar et al., 2004; Austin and Vitousek, 1998; Cheng et al., 2009; Peri et al., 2012) and global scales (Amundson et al., 2003; Craine et al., 2009; Handley et al., 1999), suggesting that N cycling is more open in dryland

ecosystems than in mesic ecosystems. The underlying explanation is when N supply is higher relative to biotic demand, more N is lost through leaching and gaseous N emissions (Austin and Vitousek, 1998), during which isotope fractionation is against the heavier isotope so that plant tissue and soil are enriched in $^{15}N$ (Robinson, 2001). However, the controls of atmospheric deposition on N cycling are often ignored in N isotope studies, in which N isotopes from atmospheric deposition and biological N fixation are assumed to be uniform over large regional scales (Bai et al., 2012; Handley et al.,

1999; Houlton and Bai, 2009). Besides, N losses in dryland ecosystems are likely dominated by gaseous loss instead of water-driven leaching (McCalley and Sparks, 2009; Peterjohn and Schlesinger, 1990). The $^{15}N$ natural abundance of total N are limited in interpreting the specific processes governing those gaseous N losses. Therefore, it seems that the measurement of total N alone are not sufficient to reveal the response of N cycling to changing precipitation, because there are multiple processes and causes contributing to the $\delta^{15}N$ variability in plant-soil systems.

Isotopes in ammonium ($NH_4^+$) and nitrate ($NO_3^-$) can serve as a proxy record for the N processes in soils because they directly respond to *in situ* processes and integrate over their characteristics (Koba et al., 2010). For example, comparing $\delta^{15}N$ values of $NH_4^+$, $NO_3^-$, and bulk soil N could reveal the relative importance of N transformation processes (such as ammonification vs. nitrification) (Koba et al., 2010; Koba et al., 1998). The dual isotope analysis of $NO_3^-$ ($^{15}N$ and $^{18}O$ of soil $NO_3^-$) provide the evidence for microbial denitrification in oceans (Sigman et al., 2009), forests (Fang et al., 2015;

Houlton et al., 2006; Wexler et al., 2014) as well as groundwater (Minet et al., 2012). In addition, $\delta^{18}O$ of $NO_3^-$ has also been used to partition microbial nitrified $NO_3^-$ from atmospheric sources because they cover different range of $\delta^{18}O$ (Böhlke et al., 1997; Brookshire et al., 2012; Kendall et al., 2007). In addition, the positive correlation or the similar pattern between N isotopes of soil available N ($NH_4^+$, $NO_3^-$, and dissolved organic N) and plant leaves have been used to explain the preferences of plant N uptake (Cheng et al., 2010; Houlton et al., 2007; Mayor et al., 2012; Takebayashi et al., 2010). With

the rapid method development recently (Lachouani et al., 2010; Liu et al., 2014; Tu et al., 2016), the analysis of isotopic values in soil $NH_4^+$ and $NO_3^-$ has the potential to elucidate the N cycling characteristics and also their controls. However, in comparison to that of bulk soil N, the $\delta^{15}N$ of both soil $NH_4^+$ and $NO_3^-$ has rarely been reported, especially in drylands.

Soil microbes constitute a major biosphere portion in terrestrial ecosystems and play key roles in regulating ecosystem functions and biogeochemical cycles (Van Der Heijden et al., 2008). Linking the soil microbial communities and N

processes is critical for evaluating the response of N transformations to climate change. However, despite of rapid development of high-throughput sequencing techniques in recent decades, there is still a great challenge for researchers to establish such linkages owing to the limitation of techniques, especially at large spatial scales (Zhou et al., 2011). Alternatively, a microarray-based metagenomics technology, Geochip, has been developed for analysis of microbial



communities (He et al., 2007; He et al., 2010b; Tu et al., 2014). This technique not only can be used to analyze the functional diversity, composition and structure of microbial communities, but also to directly reveal the linkages between microbial communities and ecosystem functions (He et al., 2007). Functional gene microarray approach has been used to examine the response of microbe-mediated N processes in different environmental conditions. For example, denitrification genes from the soils in Antarctic were found to be linked to higher soil temperatures, and $N_2$-fixation genes were linked to the presence of lichens (Yergeau et al., 2007). Research along an elevation gradient pointed out that some of denitrification genes (*nir*S and *nos*Z) were more abundant at higher elevations, and nitrification was the major process of $N_2O$ emission in the Tibetan grassland (Yang et al., 2013). The latest version, GeoChip 5.0S, contains probes covering over 144, 000 functional genes, which enables us to understand key microbially mediated biogeochemical processes (e.g. N cycling) more deeply than ever before (Cong et al., 2015; Wang et al., 2014).

In this study, we explored the effects of water availability on ecosystem-level N availability and cycling along a 3200 km transect in northern China. This natural gradients of precipitation provides an ideal system for identifying the response of soil N dynamics to water availability. In a previous study we reported a hump-shaped pattern of $\delta^{15}N$ for bulk soil N along this precipitation gradient, with a threshold at aridity index of 0.32 (mean annual precipitation of about 250 mm), showing the respective *soil microbial vs. plant* controls (Wang et al., 2014). Here, we analysed the concentrations, N isotopic composition of soil $NH_4^+$ and $NO_3^-$ (for $NO_3^-$, also oxygen (O) isotopes) as well as microbial gene abundance associated with soil N transformations. The principal objective of this study was to examine (1) the patterns of concentrations and $\delta^{15}N$ for soil $NH_4^+$ and $NO_3^-$, and (2) the patterns of gene abundance associated with microbe regulated soil N processes; (3) how did soil N cycling respond to changes in water availability along the precipitation gradient in dryland ecosystems.

## 2 Materials and methods

### 2.1 Study areas

The research was carried out along a 3200 transect across Gansu province and Inner Mongolia in northern China, covering a longitude from 87.4º E to 120.5º E and a latitude from 39.9º N to 50.1º N (Fig. 1). The climate in this area was predominantly arid and semi-arid continental. From west to east of the transect, the mean annual precipitation (MAP) increased from 36 mm to 436 mm and mean annual temperature (MAT) decreased from 9.9 ºC to -1.8 ºC (Fig. S1), and aridity index from 0.04 to 0.60 (Fig. S1). Vegetation types distributed along the transect were mainly desert, desert steppe, typical steppe and meadow steppe; three dominant grass genera and three shrub genera were *Stipa* spp., *Leymus* spp., *Cleistogenes* spp., *Nitraria* spp., *Reaumuria* spp., and *Salsola* spp.. Soil types from west to east along the transect were predominantly arid, sandy, and brown loess rich in calcium.





## 2.2 Soil sampling and sample preparation

Soil sampling was conducted from July to August in 2012, in the peak of soil N transformations. This is the same transect as
described in Wang et al. (2014), but with slightly different site coverage; we selected 36 sites at about 100 km intervals
between adjacent sites (Fig. 1). In each site, we set a 50 m × 50 m plot, and then five 1 m × 1 m subplots at the four corners
and the centre of the plot. In each subplot, twenty random mineral soil samples were collected using soil cores (2.5 cm
diameter × 10 cm depth) and thoroughly mixed together into one composite sample. The fresh soils were sieved (2 mm) to
remove roots and rocks, homogenized by hand and separated into three parts. The first part was extracted in 2 M KCl (1:5 w/
v) for 1 h on each sampling day, and the extracts were stored at -4 °C during the sampling trip. The second part was placed
into a sterile plastic bag and immediately stored at -40 °C for later DNA extraction. The third part was stored into a plastic
bag in a refrigerator at -4 °C for subsequent analyses.

## 2.3 Analyses of soil physicochemical properties and isotopes

Soil pH was measured using pH meter with 1:2.5 ratio of soil to water. Soil nitrogen content and isotopes were determined
by Elemental analyser connected to Isotope Ratio Mass Spectrometer (IRMS) (Wang et al., 2014). The concentrations of soil
$NH_4^+$ and $NO_3^-$ in KCl extracts were analysed using conventional colorimetric methods (Liu et al., 1996). Ammonium
concentrations were determined by the indophenol blue method, and nitrate by sulfanilamide-NAD reaction following
cadmium (Cd) reduction.

The analyses of isotope compositions of $NH_4^+$ and $NO_3^-$, including $\delta^{15}N$ of $NH_4^+$, $\delta^{15}N$ of $NO_3^-$, and $\delta^{18}O$ of $NO_3^-$ ($\delta =$
$[(R_{sample}/ R_{standard}) -1] \times 1000$, where R denotes the ratio of heavy isotope to light isotope for N or O, in units of per mil, ‰),
were based on the isotopes analysis of nitrous oxides ($N_2O$). Specifically, $NH_4^+$ in the extract was firstly oxidized to $NO_2^-$ by
alkaline hypobromite ($BrO^-$), and then reduced into $N_2O$ by hydroxylamine ($NH_2OH$) (Liu et al., 2014). Nitrate was initially
reduced into $NO_2^-$ by Cd power, and then reduced into $N_2O$ by sodium azide ($NaN_3$) in an acetic acid buffer (McIlvin and
Altabet, 2005; Tu et al., 2016). In order to correct machine drift and blank over the isotope analyses, the international
standards of $NH_4^+$ (IAEA N1, USGS 25, and USGS 26) and $NO_3^-$ (IAEA N3, USGS 32, and USGS 34) were treated in the
same analytical procedures as the sample to obtain the calibration curve between the measured and their expected isotope
values. The isotopic signatures of produced $N_2O$ was determined by IsoPrime 100 continuous flow isotope ratio mass
spectrometer connected to Trace Gas (TG) preconcentrator, whose setup was described in Liu et al. (2014). The analytical
precision for isotopic analyses was better than 0.3 ‰ (n = 5).

## 2.4 DNA extraction and GeoChip analysis

For soil DNA extraction, purification, quantification, and analysis of functional structure of the soil microbial communities,
we adopted the same approaches as described previously (Wang et al., 2014). In addition to the abundance of nitrification
and denitrification genes reported in Wang et al. (2014), the gene abundance of N fixation, ammonification, and anaerobic



ammonia oxidation (anammox) will be included in this study. Briefly, microbial genomic DNA was extracted from 0.5 g soil
using the MoBioPowerSoil DNA isolation kit (MoBio Laboratories, Carlsbad, CA, USA) and purified by agarose gel
electrophoresis followed by phenol-chloroform-butanol extraction as previously described. DNA quality was assessed by the
ratios of A260/280 and A260/230 using NanoDrop ND-1000 Spectrophotometer (NanoDrop Technologies Inc., Wilmington,
DE), and final soil DNA concentrations was quantified by PicoGreen using a FLUOstar Optima (BMG Labtech, Jena,
Germany). GeoChip 5.0S, manufactured by Agilent (Agilent Technologies Inc., Santa Clara, CA), was used for analyzing
DNA samples. The experiments were conducted as described previously (Wang et al., 2014). In brief, the purified DNA
samples (0.6 μg) was used for hybridization, which was labelled with the fluorescent dye Cy 3. Subsequently, the labelled
DNA was resuspended and hybridized at 67 °C in Agilent hybridization oven for 24 h. After washing and drying, the slides
were scanned by a NimbleGen MS200 scanner (Roche, Madison, WI, USA) at 633 nm using a laser power of 100 % and
photomultiplier tube gain of 75 %, respectively. The images data were extracted by Agilent Feature Extraction program. The
microarray raw data were further processed for subsequent analysis using an in-house pipeline that was built on the platform
at the Institute for Environmental Genomics, University of Oklahoma (He et al., 2010a; He et al., 2007).

### 2.5 Statistical analyses

All analyses were conducted by using the SPSS 18.0 (SPSS, Chicago, IL) for windows. Pearson correlation analyse was
conducted to examine the linear relationships among different variables. Independent-Samples T test was performed to
examine the differences in the investigated variables between arid zone soils and semiarid zone soils. Statistically significant
differences were set at a $P$-value of 0.05 unless otherwise stated.

### 3 Results

### 3.1 Soil $NO_3^-$ and $NH_4^+$ concentrations

We found significant inorganic N accumulation in the investigated soil layer (0-10 cm) in sites with MAP of less than 100
mm (Fig. 2b and c). Furthermore, the abundance of microbial gene associated with soil N transformations was significantly
lower than those in sites with MAP of more than 100 mm (Fig. 4). Together with the vegetation distribution along the
transect (Fig. 1), these results pointed out that the N status and their impacting factors could be different on the two sides of
MAP = 100 mm. For convenience, thereafter we refer the area with MAP from 36 mm to 102 mm (15 sites) and from 142
mm to 436 mm (21 sites) as the arid zone and semiarid zone in this study, respectively.
Soil $NO_3^-$ and $NH_4^+$ concentrations in the arid zone were significantly higher than those in the semiarid zone ($P < 0.001$;
Fig. 2b and c). In the arid zone, $NO_3^-$ concentrations were highly variable and up to 1400 mg N $kg^{-1}$, with a mean of 87 mg
N $kg^{-1}$ (Fig. 2c). Ammonium concentrations varied from 2.0 to 9.9 mg N $kg^{-1}$, with a mean of 4.3 mg N $kg^{-1}$ (Fig. 2b). In the
semiarid zone, $NO_3^-$ and $NH_4^+$ concentrations were low and less than 5 mg N $kg^{-1}$ in most samples. Soil $NH_4^+$ concentrations
showed a quadratic relationship with increasing MAP in the semiarid zone, but $NO_3^-$ concentrations remained low and did



not change with increasing MAP. As expected, soil total N was significantly high in the semiarid zone (on average 0.1 %) than in the arid zone (on average 0.02 %) and increased dramatically with increasing precipitation in the semiarid zone (Fig. 2a). Our results suggest a higher inorganic N availability in the arid zone than in the semiarid zone despite a smaller total N pool therein, which support the idea that N availability is relatively higher in dry areas than in less dry areas.

### 3.2 The $^{15}$N natural abundance of soil $NO_3^-$ and $NH_4^+$

The $\delta^{15}$N values of $NO_3^-$ were significantly higher in the semiarid zone (0.5 to 19.2 ‰) than in the arid zone (-1.2 to 23.4 ‰; $P < 0.01$; Fig. 2f), with the mean of 8.4 ‰ and 6.3 ‰, respectively. With increasing MAP, the $\delta^{15}$N value of $NO_3^-$ increased in the arid zone but decreased in the semiarid zone, suggesting different controlling factors in the areas with different water availability. Different from that of soil $NO_3^-$, the $\delta^{15}$N value of $NH_4^+$ was significantly higher in the arid zone (-1.2 to 20.2 ‰) than in the semiarid zone (-13.9 to 12.6 ‰; $P < 0.01$; Fig. 2e), with the mean of 9.2 ‰ and -0.3 ‰,

respectively. The $\delta^{15}$N of $NH_4^+$ was negatively correlated with MAP in the semiarid zone, but was stable as precipitation increased in the arid zone.

   The N isotopic signature of $NH_4^+$ and $NO_3^-$ reflects not only isotopic fractionation during N transformation processes, but also the N isotopic signature of their main sources, i.e., bulk soil N and $NH_4^+$, respectively. Therefore, we also calculated the relative $^{15}$N enrichment of soil $NH_4^+$ (the difference between $\delta^{15}$N of $NH_4^+$ and bulk soil N) and $NO_3^-$ (the difference

between $\delta^{15}$N of $NO_3^-$ and $NH_4^+$) to examine the isotopic imprint of N transformations on soil $NH_4^+$ and $NO_3^-$. The relative $^{15}$N enrichment of soil $NH_4^+$ in the arid zone were mostly positive, while they were negative in the semiarid zone (Fig. 3a). There was a negative correlation between MAP and the relative $^{15}$N enrichment of soil $NH_4^+$ across both arid zone and semiarid zone (Fig. 3a). According to Rayleigh model, sinks are always $^{15}$N depleted than sources (Robinson, 2001). The positive values for the $^{15}$N enrichment of soil $NH_4^+$ support that net $NH_4^+$ losses occurred mainly in the arid zone, while the

negative values imply that net $NH_4^+$ gain (via microbial and plant regulation) might increase in the semiarid zone, and subsequently reduced the relative $^{15}$N enrichment of soil $NH_4^+$. In a similar way, we found that the relative $^{15}$N enrichment of $NO_3^-$ were mostly negative in the arid zone and positive in the semiarid zone (Fig. 3b). The positive correlationship was observed between MAP and the $^{15}$N enrichment of soil $NO_3^-$ in both arid and semiarid zone (Fig. 3b). Accordingly, these results suggest that $NO_3^-$ losses along this dryland transect occurred when water becomes more available, and progressively

enriched residual soil $NO_3^-$ in $^{15}$N.

### 3.3 The abundance of microbial functional genes

The abundance of microbial gene of five main N cycling groups (N fixation, ammonification, nitrification, denitrification, and anammox (anaerobic ammonium oxidation)) was detected in study area. In arid zone soils, gene abundances of all detected N cycling groups was found to be extremely low (Fig. 4), indicating limited microbial potential in the very dry

environment. There was a sharp increase (by 8 to 9 fold) of gene abundance of all detected N cycling group from the arid zone to the semiarid zone soils (with MAP from 102 to 142 mm; Fig. 4), even though the MAP was still low and soils were





dry in the time of sampling (see soil moisture in Fig. S2). Gene abundance in the semiarid zone was 1-2 orders of magnitude higher than those in the arid zone. In addition, microbial gene abundance of five main N cycling groups all increased with increasing precipitation in both arid and semiarid zone (Fig. 4), suggesting the potential regulation of water availability on
soil microbial processes.

## 4 Discussion

### 4.1 Losses of soil $NO_3^-$ and $NH_4^+$

Water availability drives different patterns of N cycling at the two sides of about MAP = 100 mm in this transect. In the semiarid zone, the increasing precipitation seems lead to the losses of N in soil nitrate, but not in soil ammonium (Fig. 2b
and c). Nitrate can be lost via denitrification and leaching. Based on the measurement of dual isotopes ($\delta^{15}N$ and $\delta^{18}O$) of soil $NO_3^-$, we found the direct evidence for denitrification in the semiarid zone soils. Microbial denitrification was a kinetic reactions with strong isotopic fractionation from 5 to 25 ‰ for nitrate nitrogen and oxygen (Granger et al., 2008). This kind of fractionation results in concurrent increases in the $\delta^{15}N$ and $\delta^{18}O$ values of the remaining $NO_3^-$ with a ratio of 0.5 to 1 (Kendall et al., 2007). In the present study, $\delta^{18}O$ values of soil $NO_3^-$ were significantly correlated with $\delta^{15}N$ values of soil
$NO_3^-$ in the semiarid zone, with a slope of 0.7 (Fig. 5b), indicating the occurrence of denitrification driven N losses when water becomes relatively available. At the same time, the relative $^{15}N$ enrichment of soil $NO_3^-$ also pointed out that denitrification might be stronger with increasing precipitation (Fig. 3b). The increased denitrification may result from the availability of N and $O_2$ supply (Saggar et al., 2013). Enhanced nitrification rates as a result of water addition would yield a greater amount of nitrate for potential denitrification. Increased soil respiration under hot spot and/or hot moment caused by
pulse precipitation consumes more $O_2$, favouring denitrification (Abed et al., 2013). In addition, our preliminary study also showed an increasing $N_2$ losses via denitrification in the semiarid soils by using a $^{15}N$-labelled incubation experiment (unpublished data). These results support our previous finding in the same precipitation gradient that gaseous N losses is increasing as precipitation increases (Wang et al., 2014).

In the arid zone, the low microbial gene abundances suggested lower biological activities (Fig. 4). However, $\delta^{15}N$ of
soil $NO_3^-$ was increasing in the arid zone (Fig. 2f), and they were higher than those of soil $NH_4^+$ in several sites (Fig. 3b), pointing the losses of soil $NO_3^-$. Soil microbial denitrification could happen in hotspots after the heavy pulse precipitation (Abed et al., 2013; Zaady et al., 2013), resulting in the $^{15}N$-enriched soil $NO_3^-$. Alternatively, chemodenitrification may attribute to soil $NO_3^-$ losses in the arid zone. Chemodenitrification is a abiotic processes, in which the reduction of $NO_3^-$, $NO_2^-$, NO, and $N_2O$ is coupled to the oxidation of Fe (II) (Zhu-Barker et al., 2015). Because ample soil $NO_3^-$ was preserved
in the arid zone (Fig. 2c), chemodenitrification can occur when soil $NO_3^-$ contacts with metal (e.g. Fe (II)) minerals.

Different from the $\delta^{15}N$ of soil $NO_3^-$, $\delta^{15}N$ values of soil $NH_4^+$ and their relative $^{15}N$ enrichment were higher in the arid zone than those in the semiarid zone (Fig. 2e and 3a), suggesting the losses of $NH_4^+$ or nitrification in the drier sites. Because soil pH was higher in the arid zone (from 7.3 to 9.7; Fig. 6a), $NH_3$ volatilization can be strong for the $NH_4^+$ losses.



Fractionation effect of $NH_3$ volatilization was reported to be 40-60 ‰ (Robinson, 2001), thus this process can result in $^{15}N$-

enriched soil $NH_4^+$. This is further supported by the significantly negative correlation between the $\delta^{15}N$ values of $NH_4^+$ and

soil pH (Fig. 6b).

   In the semiarid zone, soil $NH_4^+$ was gradually depleted in $^{15}N$ relative to the bulk soil N (Fig. 3a), suggesting the net

$NH_4^+$ gain with increasing precipitation in this N limited areas. $NH_3$ volatilization in the semiarid zone soils likely became

less important due to relatively lower pH compared to those in the arid zone soils (Fig. 6a). Previous studies have found that

water addition did not stimulate $NH_3$ volatilization in arid ecosystems (Yahdjian and Sala, 2010). Here, we suggest that the

consumption of soil $NH_4^+$ in the semiarid zone may include following processes. First, plant uptake will be enhanced when it

is coupled with the microbe-regulating N cycling. The increased aboveground biomass with increasing MAP suggested a

potentially higher net plant N accumulation along the precipitation gradient (Wang et al., 2014). Since soil $NH_4^+$ was higher

relative to soil $NO_3^-$ in the semiarid zone ($P < 0.001$), the dominant plant species might prefer soil $NH_4^+$ to $NO_3^-$.

Furthermore, we observed a close relationship between the $\delta^{15}N$ values of plant leaves (non-N fixing species) and soil $NH_4^+$

($R^2 = 0.40$; Fig. 7a), but not for soil $NO_3^-$ (Fig. 7b). Further, when we plot this correlation for each plant species, one half of

them (for *Stipa* spp., *Cleistogenes* spp., and *Reaumuria* spp.) was significantly correlated with soil $NH_4^+$, but still not for soil

$NO_3^-$. Nevertheless, this result demonstrates the N preference of plant uptake, as suggested by previous studies (Cheng et al.,

2010; Houlton et al., 2007). Second, soil nitrification could increase as indicated by the microbial abundance associated with

nitrification genes (Fig. 4). Soil nitrification have been observed to be enhanced with more water widely (Dijkstra et al.,

2012; Yahdjian and Sala, 2010). However, we suggest that the extent of nitrification originating from soil $NH_4^+$ (autotrophic

nitrification) in the semiarid zone may be lower than soil ammonification, because this process (with a fractionation effect of

15-30 ‰) did not cause isotopic enrichment of soil $NH_4^+$. Besides, microbes might prefer $^{15}N$-enriched soil $NH_4^+$ during

microbial immobilization even in the N limited areas (Makarov et al., 2008), thus the remaining $NH_4^+$ could become $^{15}N$-

depleted. However, such assumption needs further confirmation in our study area.

   Unexpectedly, we detected anammox gene in these drylands ecosystems. The gene abundance of anammox was even

significantly higher than other N processes in both arid and semiarid zone soils ($P < 0.001$; Fig. 4). Anammox is the

microbial reaction between $NH_4^+$ and $NO_2^-$ with $N_2$ as the end product (Thamdrup and Dalsgaard, 2002). Previous studies

have found equal contributions of anammox to the consumption of both soil $NH_4^+$ and $NO_3^-$ (potential source of $NO_2^-$) in N-

loaded and water-logging areas (Yang et al., 2014; Zhu et al., 2013). However, the only two studies regarding anammox rate

so far were failed to confirm its importance in drylands (Abed et al., 2013; Strauss et al., 2012). Thus, although anammox

possesses a fractionation effect of 23-29‰ (Brunner et al., 2013), it is hard to tell the significance of anammox in our study

area at this moment.

   Other abiotic processes have also been widely reported contributing to N losses in drylands. In addition to $NH_3$

volatilization, high soil surface temperature driven by solar radiation may be responsible for the gaseous losses of N in

dryland ecosystems (Austin, 2011; McCalley and Sparks, 2009, 2008), contributing to the $^{15}N$ signature of soil mineral N.





Other non-fractionation processes might also influence N cycle in dryland ecosystems, such as aerolian deposition and water erosion (Austin, 2011; Hartley et al., 2007).

## 4.2 Sources of soil NO$_3^-$ and NH$_4^+$

We observed the high concentrations of soil nitrate in the surface soil (0-10 cm) in the arid zone of our study area (Fig. 2c), which is about 20 times higher than that in the semiarid zone on average. Nitrate can be formed via microbial nitrification or deposited from N-bearing gaseous (e.g. HNO$_3$) or dry aerosol NO$_3^-$ (Kendall et al., 2007). If nitrate formed by nitrification, nitrified NO$_3^-$ will contain the oxygen atom from soil O$_2$ and H$_2$O in a 1:2 ratio (Kendall et al., 2007). Since δ$^{18}$O value of atmospheric O$_2$ is relatively stable (23.5 ‰), the δ$^{18}$O value of nitrified NO$_3^-$ depends on δ$^{18}$O value of the local water. The

δ$^{18}$O values of rainwater taken from the areas, where is closer to the arid zone of our dryland transect (Lanzhou and its surrounding areas), were from -19.1 ‰ to 5.2 ‰ (Chen et al., 2015), yielding the δ$^{18}$O of nitrified NO$_3^-$ from -5.3 to 11.3 ‰ (Fig. 5a). However, the δ$^{18}$O values of soil NO$_3^-$ in the arid zone varied from 5.5 ‰ to 51.8 ‰ (Fig. 5a). This disparity between calculated and measured δ$^{18}$O values provides evidence for a minor importance of nitrification. In addition, approximately half of the δ$^{18}$O values of soil NO$_3^-$ were larger than atmospheric O$_2$ (23.5 ‰) in the arid zone. These higher

δ$^{18}$O values of NO$_3^-$ have rarely been reported for nitrified NO$_3^-$ (Kendall et al., 2007). An *in situ* study conducted in the forest floor soils found that δ$^{18}$O values of nitrified NO$_3^-$ were changing from 3.1 ‰ to 10.1 ‰ (Spoelstra et al., 2007). By comparison, atmospheric origin NO$_3^-$ normally have relatively higher δ$^{18}$O values because of the chemical oxidation of nitrate precursor, NO$_x$ (NO and NO$_2$) (Fang et al., 2011). For example, previous research found that δ$^{18}$O values of nitrate aerosol are from 60 ‰ to 111 ‰ in the Dry Valleys of Antarctica (Savarino et al., 2007). These information together

supports the hypothesis that a fraction of nitrate in the surface soils of the arid zone were resulted from the atmospheric deposition. Nitrate would be accumulated on the surface soil when experiencing prolong droughts, as have been found in northern Chile and southern California (Böhlke et al., 1997), and in the Turpan-Hami area of northwestern China (Qin et al., 2012). A pronounced trend (green arrow; Fig. 5a) toward higher δ$^{18}$O and lower δ$^{15}$N values is obvious for elevated NO$_3^-$ concentrations, which might be the mixed NO$_3^-$ from soil nitrification and the accumulation of atmospheric origin, as

previously observed in groundwater of Saharan desert (Dietzel et al., 2014). In the arid zone of our study area, extreme dryness and high alkalinity (with a pH average of 8.3) might limit microbial activities, as suggested by the low gene abundance involved N transformation processes (Fig. 4). The decreased microbial activities, together with the low plant N uptake due to small plant cover (Fig. 1), as reported previously (Zhu et al., 2006), would facilitate the preservation of NO$_3^-$ in the soil surface. In addition, different from previous finding that subsoil nitrate accumulated in desert soils (Walvoord et

al., 2003), the high nitrate concentration in the surface soil probably indicated that leaching was not important in the arid zone soils of our study area.

     In the semiarid zone, the δ$^{18}$O values of soil NO$_3^-$ were low (0.9-21.0 ‰), indicating much less atmospheric contribution. The deposited NO$_3^-$ will experience postdepositional processes via microbes, and thus the signature of δ$^{18}$O of atmospheric NO$_3^-$ faded after a period of time (Qin et al., 2012). With the increasing of MAP, nitrification progressively





provided more $NO_3^-$ with lower $\delta^{18}O$ values, reflecting isotopic signatures of soil $O_2$ and $H_2O$. The calculated $\delta^{18}O$ of nitrified $NO_3^-$ was from 2.5 to 6.5 ‰ based on the $\delta^{18}O$ of soil $H_2O$ (-8 to -2 ‰; Shenyang site) (Liu et al., 2010). It is coincidence that the difference (-3.6-16.5 ‰) between the calculated $\delta^{18}O$ of nitrified $NO_3^-$ and measured $\delta^{18}O$ of soil $NO_3^-$ in the semiarid zone soils overlapped with the fractionation effect of denitrification for nitrate oxygen (5-25 ‰) (Granger et al., 2008). In addition, as the gradually $^{15}N$-depleted soil $NH_4^+$ in the semiarid zone, we suggested that heterotrophic

nitrification can be important in contributing to soil $NO_3^-$ pool. Soil heterotrophic nitrification is the process that oxidize organic N to $NO_3^-$. It does not consume $NH_4^+$, thus might not result in $^{15}N$-enriched in soil $NH_4^+$ (Fig. 3a). Moreover, the importance of heterotrophic nitrification have been recognized by recent investigations in permanent grasslands (Müller et al., 2014; Müller et al., 2004) and forests (Zhang et al., 2014).

There was also a slight $NH_4^+$ accumulation in soils of the arid zone (Fig. 2b). This might be the combined results of

many biotic and abiotic processes including $NH_3$ volatilization mentioned above and atmospheric deposition. Ammonium have been shown to be the dominant species in atmospheric deposition in China (Liu et al., 2013), thus it could be one of those processes contributing to $NH_4^+$ accumulation in the arid zone. Furthermore, we observed higher $\delta^{15}N$ value in aerosol $NH_4^+$ (up to 40 ‰ in summer; unpublished data), which may partially explain the $^{15}N$-enriched soil $NH_4^+$ in the arid zone (Fig. 2e). Besides, with the exception of the biological N fixation by legume plant (*Caragana* spp.) showed in the same

transect (Wang et al., 2014), in this study, we speculated that biological N fixation by BSCs (Wu et al., 2009; Zhuang et al., 2015) and soil ammonification also contributed to soil $NH_4^+$ pool in the arid zone. The former process provided $NH_4^+$ with $\delta^{15}N$ value around zero and the later provided $NH_4^+$ with $\delta^{15}N$ value slightly lower than those of soil organic nitrogen.

In the semiarid zone with MAP from 100 mm to 200 mm, soil $NH_4^+$ concentrations were lower than those in the arid zone, which may be caused by a tight coupling between $NH_4^+$ production and consumption processes (Nielsen and Ball,

2015). Soil $NH_4^+$ was depleted in $^{15}N$ relative to bulk soil N and their differences in $\delta^{15}N$ increased with increasing MAP (Fig. 3a). This result might reflect isotopic fractionation during ammonification as observed in previous study (Koba et al., 2010; Robinson, 2001; Zhang et al., 2015), at the same time ammonification was stimulated with increasing MAP. We also suggested that, with higher water availability, N turnover linking plant N uptake and return would have enhanced and progressively fuelled soil ammonification, which in turn result in lower $\delta^{15}N$ in soil $NH_4^+$. The increasing $NH_4^+$

concentrations in semiarid zone seems to agree with our assumption that, in addition to autotrophic nitrification, heterotrophic nitrification is relevant in our study area, and thus the combined nitrification process do not consume soil $NH_4^+$ too much. In addition, there is also a possibility of dissimilatory nitrate reduction to ammonium (DNRA) although we did not measure this process in our study. DNRA is even less sensitive to oxygen level than denitrification and therefore may also occur in aerobic soils (Müller et al., 2004), contributing to the availability of soil $NH_4^+$.




## 5 Summary


To the best of our knowledge, our study for the first time showed the pattern of $\delta^{15}N$ in soil inorganic N ($NH_4^+$ and $NO_3^-$) across a large precipitation range in drylands. Together with the analysis of soil N concentration, soil properties, and functional gene abundance, the results here demonstrate the shifting contribution of *abiotic vs. biotic* (microbes and plants) controls on N cycling along this 3200 km dryland transect with a threshold at MAP of around 100 mm.

350       In the arid zone with extreme aridity (36 mm < MAP < 100 mm; Fig. 8a), plant cover is sparse and microbial activity is limited (Fig. 1 and 4). Nitrogen input, mostly in the form of atmospheric deposition, is largely accumulated, creating "enriched" inorganic N pools despite a much smaller pool of soil total N. The accumulation of inorganic N drives abiotic processes that lead to N losses with strong isotope fractionation effect on the remaining soil N. Higher pH associated with lower MAP is likely a dominant driver of $NH_3$ volatilization, causing soil $NH_4^+$ enriched in $^{15}N$. Studies have reported

phytochemical $NO_3^-$ loss in desert soils. However, the very high yet variable $NO_3^-$ accumulation in soil comparing to $NH_4^+$ suggests limited $NO_3^-$ loss under extreme aridity.

      In the semiarid zone (100 mm < MAP < 436 mm; Fig. 8b), controls on N cycling increasingly shift from abiological to biological factors. Microbial gene abundances associated with N cycling groups were 1-2 orders of magnitude higher than those in the arid zone (Fig. 3). Increasing ammonification with increasing MAP both reduced $NH_3$ volatilization (with lower

pH) and provided lighter N isotope for soil $NH_4^+$. Higher ammonification (N mineralization) would favour both plant uptake and nitrification. Denitrification could lead to $NO_3^-$ loss because of less preference of $NO_3^-$ by plant uptake, meanwhile soil heterogeneity and pulse precipitation could provide hotspots for microbial organisms performing this process. The abiotic vs. biotic controls on N cycling and N losses around a threshold of MAP 100 mm suggest impacts of global climate changes, in particular the change of precipitation pattern, on these dryland ecosystems.

**Author contribution**

Y. Fang, D. Liu, W. Zhu, and X. Han designed the study; D. Liu, X. Wang, Y. Pan, C. Wang, D. Xi, Y. Wang, and X. Han performed the experiment; D. Liu, W. Zhu, Y. Fang, X. Wang, Y. Pan, C. Wang, D. Xi, E. Bai and Y. Wang analysed the data. D. Liu, W. Zhu, and Y. Fang wrote the manuscript; X. Wang, Y. Pan, C. Wang, E. Bai, and X. Han contributed to discussion of the results and manuscript preparation.

**Acknowledgements**

The work was financially supported by the Strategic Priority Research Program of the Chinese Academy of Sciences (XDB15020200, XDB15010401 and XDA05100100), the National Natural Science Foundation of China (31370464, 31422009 and 41405144), Hundred Talents Program of Chinese Academy of Sciences (No.Y1SRC111J6), and State Key Laboratory of Forest and Soil Ecology (LFSE2015-19). We would like to thank Ying Tu, Haiyan Ren, Shasha Zhang, Feifei



Zhu, and Xiaoming Fang for their assistance in field sampling and laboratory analysis. We thank Shaonan Huang for sharing the unpublished data. We also thank all members of the sampling team from the Institute of Applied Ecology, Chinese Academy of Sciences for their assistance during field sampling.

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



**Figure captions**

**Figure 1.** Vegetation types and sampling sites distribution along the transect. Across the 3200 km precipitation gradient in northern China, four typical vegetation types are distributed from west to east, which are desert (a), desert steppe (b), typical steppe (c), and meadow steppe (d), and the dominant plant genera change gradually from shrub (*Nitraria* spp., *Reaumuria* spp., and *Salsola* spp.) to perennial grasses (*Stipa* spp., *Leymus* spp., *Cleistogenes* spp.). Soil types are predominantly arid, sandy, and brown loess rich in calcium from west to east of the transect. A total of 36 soil sampling sites were selected.

**Figure 2.** Nitrogen concentrations and isotopic composition of bulk soil N, $NH_4^+$, and $NO_3^-$. The significant ($P < 0.05$) trends are shown with a regression line (red) and 95% confidence intervals (blue). In each site, n=5.

**Figure 3.** The relative $^{15}N$ enrichment of soil $NH_4^+$ and $NO_3^-$. The relative $^{15}N$ enrichment of soil $NH_4^+$ and $NO_3^-$ were calculated as the difference between $\delta^{15}N$ of bulk soil N and $NH_4^+$, and between $\delta^{15}N$ of soil $NH_4^+$ and $NO_3^-$, respectively. The significant ($P < 0.05$) trend is shown with a regression line (red) and 95% confidence intervals (blue). In each site, n = 5.

**Figure 4.** Changes in the abundance of microbial gene involved in N cycling. Signal intensity was standardized based on both the number of array probes and DNA quantity in a gram of dry soil. Data are the site-averaged value; results of the abundance of nitrification and denitrification genes have been reported in a previous study (Wang et al., 2014). The significant ($P < 0.05$) trends are shown with a regression line (red) and 95% confidence intervals (blue).

**Figure 5.** Relationship between $\delta^{18}O$ and $\delta^{15}N$ of soil $NO_3^-$. The range of $\delta^{18}O$ and $\delta^{15}N$ from atmospheric $NO_3^-$ was based on the limited isotope measurement of precipitation. Black points represent precipitation $NO_3^-$ collected from an urban site in Beijing in the year of 2012, with data derived from Tu et al. (2016). Grey points represent precipitation $NO_3^-$ collected from Qingyuan forest CERN (Chinese Ecosystem Research Network, CERN) in Northern China in the year of 2014 (Huang et al.; unpublished data). The range of $\delta^{15}N$ and $\delta^{18}O$ produced by nitrified $NO_3^-$ are positioned by using the $\delta^{15}N$ of soil $NH_4^+$ in this study (Fig. 2e), and the estimated $\delta^{18}O$ from soil nitrification based on the 1:2 ratio of soil $O_2$ and $H_2O$ (see Text), respectively.

**Figure 6.** Soil pH and the relationship with $\delta^{15}N$ of soil $NH_4^+$. The different patterns of soil pH was observed above and below the threshold at MAP of about 100 mm; data were derived from Wang et al. (2014). There was a positive correlation between $\delta^{15}N$ of soil $NH_4^+$ and pH across the transect. The significant ($P < 0.05$) trend is shown with a regression line (red) and 95% confidence intervals (blue). In each site, n = 5.



**Figure 7.** Relationship between the $\delta^{15}N$ of foliage and $\delta^{15}N$ of soil $NH_4^+$ and $NO_3^-$. Data on foliar $\delta^{15}N$ (*Stipa* spp., *Leymus* spp., *Cleistogenes* spp., *Reaumuria* spp., and *Salsola* spp.) were from the previous study of Wang et al. (2014). Almost all dominant plants were found in the area with MAP more than 100 mm (semiarid zone). Data are the site-averaged value. The significant ($P < 0.05$) trend is shown with a regression line (thick) and 95% confidence intervals (thin).

**Figure 8.** A framework of N biogeochemical cycling in dryland ecosystems in northern China. Width of arrows and size of boxes indicate the relative importance of soil N processes and pools between the arid zone (a) and semiarid zone (b). The mean pool sizes (g N m$^{-2}$) of each soil N pool based on the soil density of top 10 cm were present in the brackets.



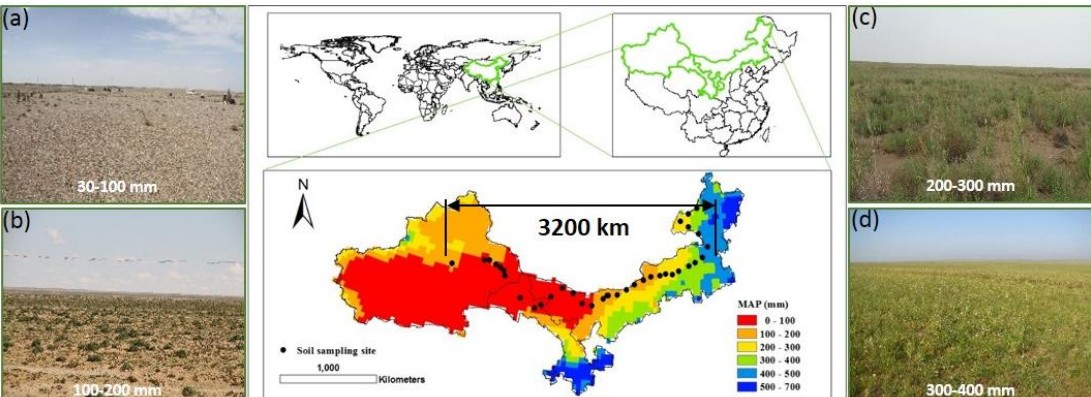

**Figure 1**





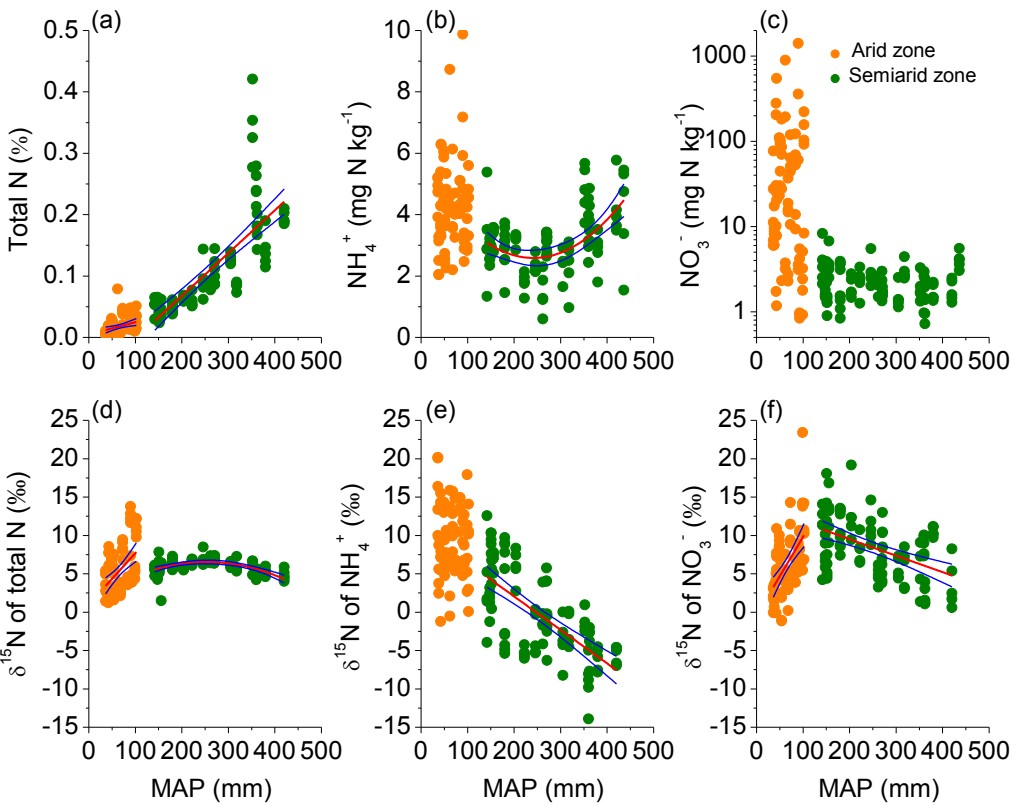

**Figure 2**





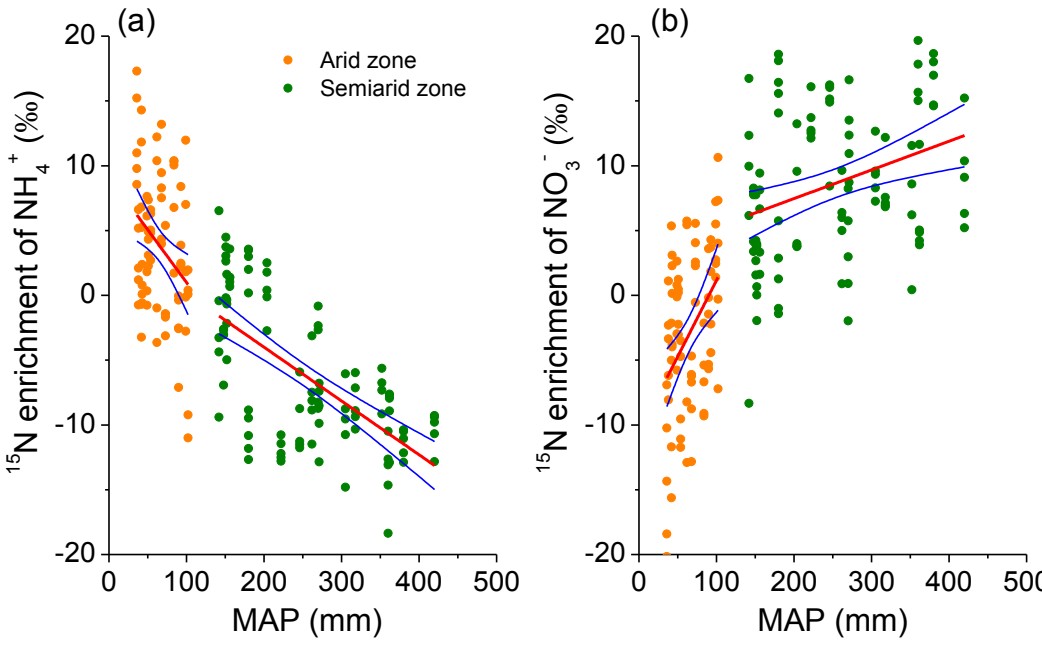


**Figure 3**





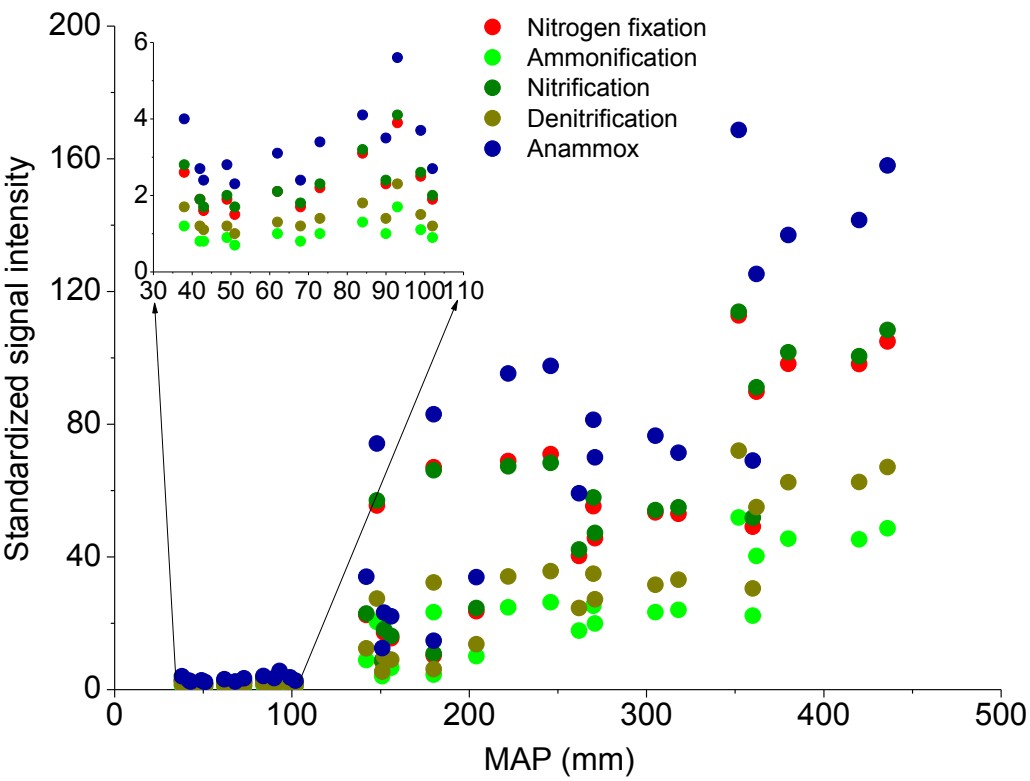

**Figure 4**




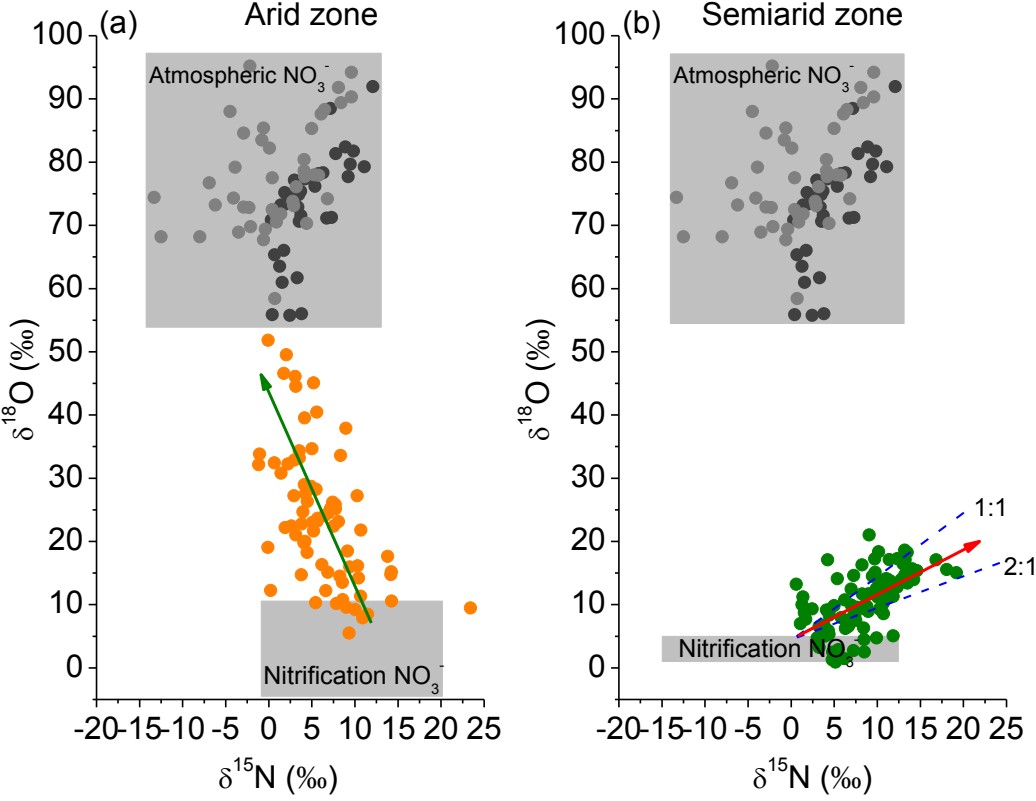

**Figure 5**





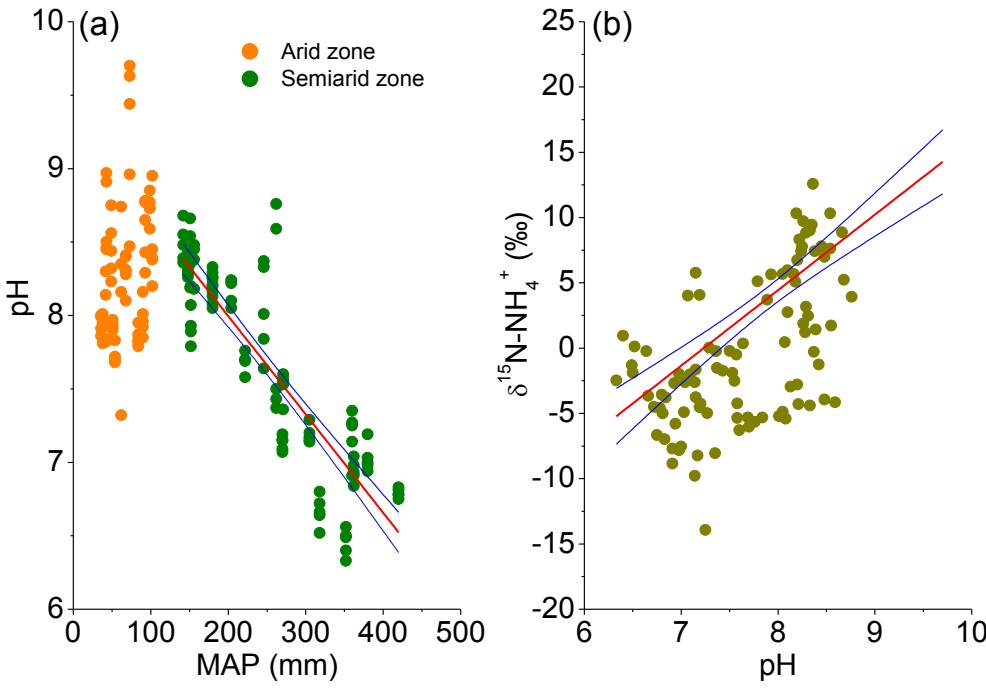


**Figure 6**



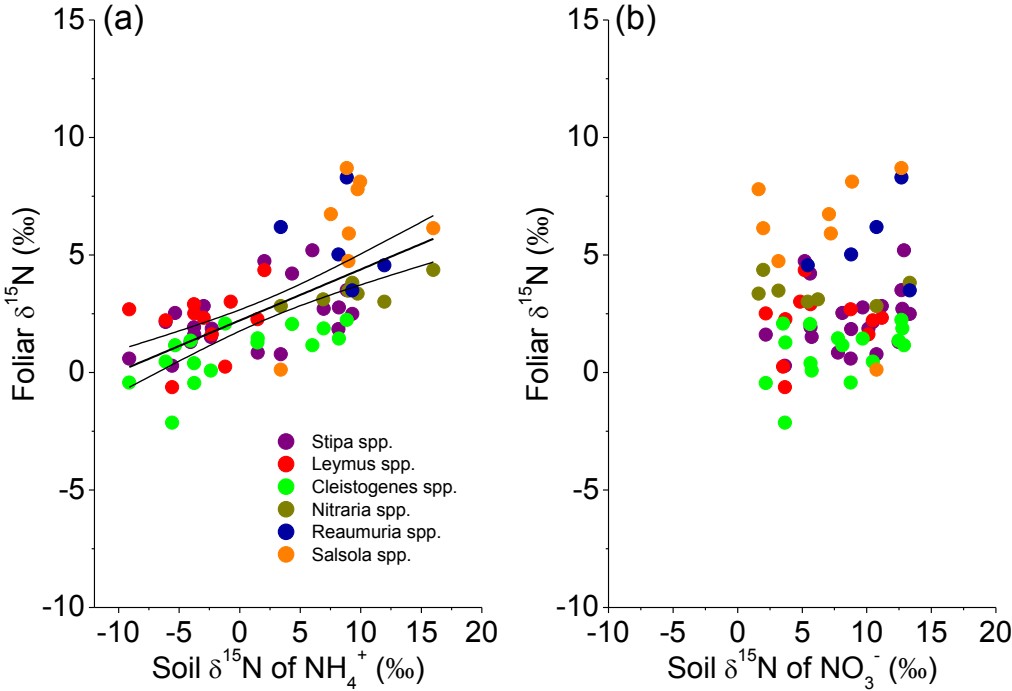

**Figure 7**





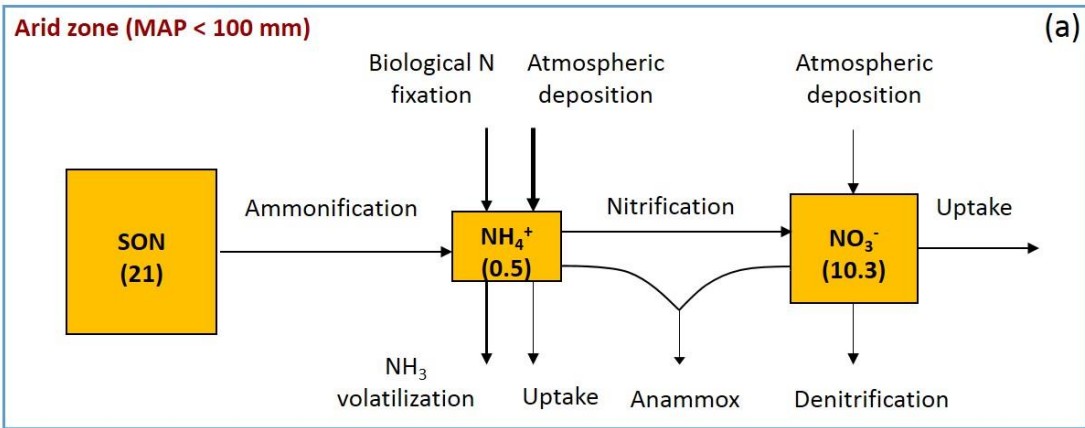

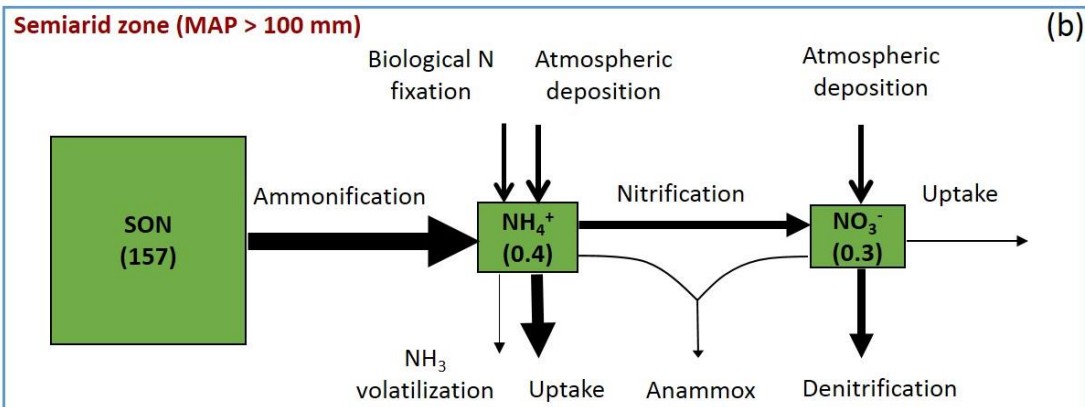

**Figure 8**
