# Peer review of "Abiotic versus biotic controls on soil nitrogen cycling in drylands along a 3200-km transect"

_Biogeosciences, 2016_

## Referee Comment (RC1) · Anonymous Referee #1 · 15 Jun 2016

General comments:

Overall, this manuscript provides an insightful data set that I believe will be of interest to anyone interested in aridland biogeochemistry. The large geographic scale and compound-specific isotopic analysis are especially valuable and the conclusions reached seem valid. Generally, the discussion of the mechanisms driving the observed patterns is thorough. There are some issues that need addressing, primarily in the discussion where several important processes have not been raised (mainly NO production), and there are several mechanisms that do not make sense (perhaps partly as a result of unclear English). This section would benefit from revision.

Generally, the figures are clear and informative. Editing for English language is necessary prior to publication.

Specific comments:

Methods: Is it just a coincidence that there is a gap in sampling sites around 100 mm MAP?

Discussion: What about loss of NO during nitrification? NO can be the dominant trace gas emitted from arid soils, and would explain loss of ammonium without subsequent appearance of nitrate. For process see Firestone & Davidson (1989) For arid land NO production see Homyak et al (2016, PNAS) and Soper et al (2016, Global Biogeochemical Cycles). This process belongs on Figure 8! A discussion of the isotopic consequences of this process should also be included.

Re foliar 15N reflecting 15N of NH4+ in the soil- this could also reflect shifting plant physiology across the significant precipitation gradient, rather than just plant source preference for ammonium. Many aspects of plant internal N cycling likely shift as a function of water availability and would influence foliar 15N. This should at least be acknowledged as an alternative explanation. Also, ammonium shows a larger range of isotopic values along the transect than nitrate, making it less likely that plant 15N would correlate with nitrate 15N anyway.

Deposition- I think you need to be clear about the difference between wet versus deposition (with different isotopic signeratures) and how you might expect them to change along the transect. Are there any measures of deposition anywhere on the transect you could mention?

Line 225- nitrate could also be removed from the soil by biological uptake. This also has potential to be a fractionating process (although evidence for fractionation by directly by plants under field conditions is limited, mycorrhizal fractionation is likely). You posit later that plant uptake of nitrate is low, but this may not necessarily be the case.

Line 235- increasing compared to what? This is important.

Line 244- There are several potential mechanisms for chemodenitrification (see again

Homyak 2016)

Line 255- Soper et al (2016, Global Biogeochemical Cycles) did find increased NH3 flux with wetting in an arid system.

Line 256- "First, plant uptake will be enhanced when it is coupled with the microbe-regulating N cycling"- I'm not sure what this means

Line 268- This doesn't make sense. I don't know of any evidence showing preference for enriched substrates- I would expect it to be exactly the reverse in fact.

Line 324- This paragraph should be rewritten for clarity- right now it's just listing off a bunch of processes and it's confusing. What are the processes that would explain more NH4+, with higher enrichment, at low precipitation? Increasing volatilization with precip explains the concentration gradient, but would induce the opposite isotopic pattern (though it depends really on how much volatilization occurs as a fraction of the standing pool). A greater proportion of atmospheric deposition versus mineralization at low precip might explain the higher 15N. If you invoke fixation by BSCs at low precip, this would also tend to decrease, rather than increase, the 15N at those sites. This also this needs to be clarified- are there BSCs on the transect? However it looks from the gene abundance data like Nfix genes increase along the transect. Rethink this paragraph.

Line 336- Does fractionation during mineralization actually increase with mineralization rate though? I don't recall any evidence for this. Also I think invoking heterotrophic nitrification, when as far as I know there isn't a lot of evidence this is an important process in the field, is a stretch. Maybe remove.

Line 359- "Increasing ammonification with increasing MAP both reduced NH3 volatilization" Why would more mineralization reduce volatilization? Unless you mean that volatilization decreases with precip? Again, I'm not sure that this is necessarily true. pH probably the main driver.

Line 360- why does more mineralization mean more plant uptake? Plant uptake is likely

more a function of water availability. These things likely co-occur, but it's not causal. Perhaps misinterpretation of wording- re-write

Figure 8- I think it needs to be clear that the size of arrows indicates qualitative interpretation of these fluxes rather than actual measurement- the presence of pool sizes on the boxes makes this especially necessary. And again, NO loss is likely much more important than ammamox and should appear here.

Technical corrections:

The manuscript contains many examples of awkward or technically incorrect English that can obscure meaning and requires editing by a native English speaker before publication. E.g. in the abstract –'our understanding of' might replace 'understanding about' and 'nitrogen cycling in drylands' rather than 'nitrogen cycling of drylands'. Also 'the patterns and mechanisms of water availability on soil N cycling' doesn't make technical sense. 'Driving' rather than 'driven'. 'Above and below' rather than 'on the two sides of'. 'Preference for' rather than 'preference of', etc.

---

## Referee Comment (RC2) · Anonymous Referee #2 · 8 Jul 2016

Review Biogeosciences Discussion BG-2016-226

The paper "Abiotic versus biotic controls on soil nitrogen cycling in drylands along a 3200 km transect" provides a great dataset on soil N cycling across a precipitation gradient in dryland ecosystems in China, based on the natural 15N (18O) abundances of bulk soils and ammonium and nitrate, and on the abundances of marker genes involved in N cycling. These novel data allow deep and unprecedented insights into the controls of inorganic N cycling of these ecosystems, and clear trends emerge in abiotic versus biotic controls.

The paper therefore addresses relevant questions within the scope of Biogeosciences. Methods and assumptions are valid, and the results definitely sufficient to support the interpretations and implications raised by the authors. The description of Materials and

methods and calculations are sufficiently complete. The authors cited relevant work and demonstrate their novel contribution to the field. The title is concise and reflects the content of work, and the abstract concise and complete in summarizing the main points of this study. The presentation/manuscript is well structured and clear, but the language should be edited by a native speaker. The number and quality of references is fair and appropriate, and supplementary material is of high quality and appropriate.

Beyond that I have the following comments (according to the lines in the manuscript, the language corrections are by far not complete):

36: should read "driving" not driven 39: delete significantly 41: rewrite "the uptake preference for soil..." 42: soil nitrate loss could also occur by hydrological pathways (leaching) during heavy rain storms. 42: rewrite "our study suggests that the shift from abiotic..." 51: rewrite "factor" not factors 54: rewrite "still lack a full understanding of the..." 61: rewrite "over large scales" 67: change "are" to "become" 71: change "water-driven" to "hydrological losses by leaching" 73: change to "...alone is not.." 74: change to "processes that contribute" 77: what is the meaning of "integrate over their characteristics"? please be more concise. 79: rewrite "..provided evidence for..." 81: "they cover a different range" 83/84: rewrite "..to study the preferences for plant N uptake" 105: change to "gradient" 109: "gene abundances" 111/112: "with microbially regulated soil processes; and 3) how does soil N cycling..." 116: "the climate is..." 118: define the aridity index here 120: "...the three ..." 124: how do the authors decide which is the peak of soil N transformations? Is that peak vegetation season? Or the short season where the majority of rainfall occurs? Please be more specific here. 131: correct "into" to "in", twice. 134: "using a pH meter" 141: "based on the isotopic analysis of nitrous oxide" 142/143: change "into" to "to", three times. 146: rewrite "samples" 148: change "to a Trace..." 168: change to "Pearson correlation analysis" 174: it should be "at" not "in" sites 175: "genes" 177: rewrite "that the soil N status and its controls could be different..." 185: "was significantly higher...". By the way if I get the numbers correct in the arid zone bulk soil N (soil total N) would be 200 mg N/kg, with nitrate 87 mg

N/kg and ammonium 4 mg N/kg, i.e. inorganic N would on average contribute 46% to soil total N, and only 54% on average is bound as organic N in humus? 188: "supports" 203: "15N depleted relative to their sources" 205: please specify what you mean with "via microbial and plant regulation". 15N depletion of soil ammonium or less 15N enrichment can arise from microbial N mineralization (if this process exerts significant N isotope fractionation) or biological N fixation (causing inputs of N with d15N around 0 to -2 permil). ;Maybe also atmospheric ammonium/ammonia deposition. 207: rewrite "A positive correlation was..." 212: "genes" 213: "rewrite "was measured at all sites" 214: "were found to be ..." 215 "in the gene abundance of all detected N cycling groups" 217: "dry at the time...". "gene abundances in the semiarid zone were..." 218: "gene abundances of the five ..." 219: "potential control of water availability on soil microbial N processes" 223: "water availability drives different patterns" is not meaningful. Please rephrase. 223: "at both sides of about MAP = 100 mm" is really not the best phrasing, maybe rather "above and below a MAP threshold of 100 mm" 224: "seems to lead to losses of N..." 226: "we found direct evidence" 226/227: of course denitrification is a kinetic process. So what? Simply say that denitrification exerts isotope fractionation against the isotopically heavier compounds, ranging between 5 and 25 permil..." 232/233: please specify this sentence on availability of N and O2 supply – to me the meaning is not clear. 235: "in addition, a preliminary study.... an increasing N2 loss via..." 240: "in some sites,..... pointing to losses of soil ..." 241: "after heavy precipitation events" 239-245: the main pattern for soil nitrate at the arid sites is 15N depletion of nitrate relative to ammonium. Only a few sites had more positive d15N values in nitrate compared to ammonium. The explanation by enhanced denitrification during heavy rain or chemodenitrification is therefore ionly secondary. The main pattern has to be explained – why is soil nitrate 15N depleted relative to ammonium. My best guess is its production through nitrification which causes ammonium to become 15N enriched and nitrate 15N depleted (this is also an alternative explanation for the 15N enrichment of ammonium at many arid sites). I also would not expect large amounts of reduced iron (FeII) to be present at arid sites. Only in some places denitrification may also play a role, where nitrate was 15N enriched relative to ammonium. Another input of nitrate is atmospheric deposition, but its isotopic composition for that region is most probably unknown (Fig 5(a) indicates that atmospheric nitrate lies between 0 and 5 permil). 243: "chemodentrification is an abiotic process.." 244: change "preserveed" to "present" 247: "suggesting losses of . . ." 248: what is the meaning of "ammonia volatilization can be strong for the ammonium loss"??? 249: "The isotope effect of . . ." 250 "significant negative. . ." 250/251: the alternate explanation is that nitrification can also cause 15N enrichment of ammonium, and 15N depleted nitrate in many arid soils actually point to a significant role of this process, aside of ammonia volatilization. 252/253: what does "suggesting the net ammonium gain" mean? Please rephrase. 252: soil ammonium "became" gradually 15N depleted relative to. . . 252-270: the main pattern of spoil ammonium is that it becomes 15N depleted with higher MAP in semiarid sites. This CANNOT be explained with consumption processes such as plant uptake and nitrification, as in both cases (plants and nitrifiers) exert an isotope effect meaning that plants or nitrate become 15N depleted and soil ammonium 15N enriched. An inverse isotope effect has never been shown for any biological process involved in the (production) consumption of ammonium. Lines 268-270 therefore are wrong because microbes will not prefer 15N enriched ammonium during immobilization. The whole paragraph therefore is misleading and has to be rewritten. The explanation can therefore only come from 15N depleted N inputs (biological N fixation, 0 to -2 permil; atmospheric ammonium/ammonia deposition, isotope range for the region unknown?) or its production through mineralization of organic N. Though the isotope effect of N mineralization is most often said to be low or negligible, it might be high if one looks at enzymes and their isotope effects that are most likely involved in deamination of organic N forms in cells (they can be as high as 20 permil). Please consult the respective N isotope reviews such as Werner and Schmidt Phytochemistry 61 (2002) 465–484. 259: "prefer soil ammonium over nitrate" 263: "demonstrates the ammonium preference of plants 265: sentence is meaningless – "soil nitrification have been observed to be enhanced with more water widely. . ."? 271: "we detected anammox

genes in these dryland ecosystems" 275: "water-logged" 275/276: "studies of anammox process rates so far failed to..." 280: "responsible for gaseous losses..." 282: "aeolian" 285: "observed the highest..." 287: besides small deposition as dissolved nitrate in rainwater or snow. 288: "since the d18O ..." 289: "depends on the d18O..." 290: "from the areas closest to..." 291: "ranged from ... to ..." 294: I don't understand the reasoning behind this sentence, why is atm. O2 and its d18O important. It is not directly expressed in the d18O of NO3- formed in the atmosphere because this is more 18O enriched. So....? 285-311: as said before there is also evidence for nitrification in the data set, as in many arid soils nitrate is 15N depleted relative to ammonium, which indicates nitrification also to contribute to soil nitrate accumulation, aside of atmospheric deposition. There are several typos in this paragraph. 316-318: what does this coincidence of KIE denitrification and d18O of nitrate mean? This is totally disconnected. Delete. 319-323: the gradual 15N depletion of ammonium in itself, but also relative to soil total N indicates that mineralization is the main input process of soil ammonium, and that N mineralization causes 15N fractionation. Obviously nitrification also occurs, but as long as only a small fraction (like 10-20%) of soil ammonium is oxidized by autotrophic nitrifiers ammonium would still be 15N depleted relative to bulk soil. Heterotrophic nitrification is another explanation, as stated by the authors. 337: why do the authors believe that soil ammonification was stimulated with higher MAP? Where is the evidence for that? Only the ammonium concentrations? 347: was the precipitation range really large? 355: what is phytochemical nitrate loss? 360: what is "provided lighter N isotope for soil ammonium? And as this sentence states "increasing ammonification reduced ammonia volatilization". How should that happen?

---

## Author Comment (AC1) · 1 Sep 2016

Dear editor,

We would like to extend our grateful thanks to reviewer1 for his/her constructive comments and suggestions to our manuscript. In the revised version of this manuscript, we will rethink our data carefully and fix many confusing sentences that the reviewer had pointed out. In addition, we will also make further efforts to improve the English writing (including taking specific comments from another reviewer) by asking a native speaker to correct our revised manuscript before resubmission. Please find our line-by-line response to the review comments below.

General comments

Overall, this manuscript provides an insightful data set that I believe will be of interest to anyone interested in aridland biogeochemistry. The large geographic scale and compound-specific isotopic analysis are especially valuable and the conclusions reached seem valid. Generally, the discussion of the mechanisms driving the observed patterns is thorough. There are some issues that need addressing, primarily in the discussion where several important processes have not been raised (mainly NO production), and there are several mechanisms that do not make sense (perhaps partly as a result of unclear English). This section would benefit from revision. Generally, the figures are clear and informative. Editing for English language is necessary prior to publication.

Reply: Thank you very much for your appreciation of our study. The comments from the reviewer are really helpful. In the revised version of the manuscript, we will rewrite part of discussion section as suggested by the reviewer. We will improve general writing by asking a native speaker to correct our revised manuscript before submission.

Specific comments:

1) Methods: Is it just a coincidence that there is a gap in sampling sites around 100 mm MAP?

Reply: You are correct that it is just a coincidence that a gap of MAP seems appear around 100 mm. Our sampling sites were well-distributed at the distance of about 100 km between two adjacent sites. The threshold happened between site 15 (MAP = 102 mm) and site 16 (MAP = 142 mm). Please see Figure 1 in the main text. The "unintended gap" serves nicely for us to break the entire gradient into 1) arid zone and 2) semiarid zone for data synthesis and discussion.

2) Discussion: What about loss of NO during nitrification? NO can be the dominant trace gas emitted from arid soils, and would explain loss of ammonium without subsequent appearance of nitrate. For process see Firestone Davidson (1989) For arid land NO production see Homyak et al (2016, PNAS) and Soper et al (2016, Global Biogeochemical Cycles). This process belongs on Figure 8! A discussion of the isotopic consequences of this process should also be included.

Reply: Thank you for your suggestion. We agree with you that NO emission during N transformation is really important in arid soils. In the revised manuscript, we will add details about this NO losses in the section of 'The losses of nitrate and ammonium'. For example, 'The concurrence of abiotic and biotic processes governed gaseous N efflux, contributing to the remove and 15N enrichment of soil inorganic N; for example, studies had reported that nitric oxide (NO) was the dominate emissions in some drylands (Homyak et al., 2016; Soper et al., 2016). Chemodenitrification is an abiotic process, in which the reduction of $NO_3^-$ or $NO_2^-$ to NO and $N_2O$ is coupled to the oxidation of reduced metals (e.g. Fe (II)) and humic substances (Medinets et al., 2015; Zhu-Barker et al., 2015). Ample soil $NO_3^-$ was present in some arid zone soils in this study (Fig. 2c), meanwhile our companion work also observed higher available Fe in arid zone soils (Luo et al., 2016). Chemodenitrification therefore can occur when soil $NO_3^-$ contacts with metal (e.g. Fe (II)) minerals (Zhu-Barker et al., 2015). In addition, the most important reaction of chemodenitrification is the formation of NO via nitrous acid ($HNO_2$ (aqueous phase), HONO (gas phase)) decomposition (Medinets et al., 2015). In the dry soils, nitrifiers can remain active in thin water films and results in higher potential nitrification rates when conditions are right, e.g., after pulse of rain (Sullivan et al., 2012). Nitrite originated from soil nitrification may be more likely to be decomposed to NO via chemodenitrification than oxidized to nitrate (Homyak et al., 2016), explaining the loss and 15N enrichment in soil ammonium. Alternatively, nitrifier denitrification can also serve as an important mechanism for NO emission by the reduction of soil $NO_2^-$ (Homyak et al., 2016).'

In Figure 8, we will also try to compare and contrast the major soil N pools and N transformations (ammonification, nitrification, denitrification, among others) between mainly abiotic driven arid zone soils and biotic dominated semi-arid zone soils. Specific losses of N, including NO, $N_2O$, and $N_2$, are not shown; instead, discussed in the text.

3)The foliar 15N reflecting 15N of NH4+ in the soil- this could also reflect shifting plant physiology across the significant precipitation gradient, rather than just plant source preference for ammonium. Many aspects of plant internal N cycling likely shift as a function of water availability and would influence foliar 15N. This should at least be acknowledged as an alternative explanation. Also, ammonium shows a larger range of isotopic values along the transect than nitrate, making it less likely that plant 15N would correlate with nitrate 15N anyway.

Reply: Thank you for your suggestion. In the revised manuscript, we will add plant physiology as an alternative reason for the changing pattern of foliar $\delta$15N. Seen from the data of foliar 15N and inorganic 15N, foliar 15N range (-2.1 8.7‰ was basically smaller than that of ammonium (-9 16‰ and nitrate (1.6 13.3‰. Significant 15N range for soil NO3- did exist but did not correlate to the plant 15N.

4) Deposition- I think you need to be clear about the difference between wet versus dry deposition (with different isotopic signatures) and how you might expect them to change along the transect. Are there any measures of deposition anywhere on the transect you could mention?

Reply: Good point. In our companion work along this transect (Wang et al. 2014), we reported that rates of bulk N deposition (wet + dry) were increasing from west to east along this transect; data were estimated from a published paper (Lelieveld and Dentener 2000). We expect that dry deposition could be higher than wet deposition in the arid zone soils, and that in the semiarid zone soils the contribution of wet deposition will increase significantly. In previous studies, higher $\delta$15N values in dry deposition than in wet deposition had been reported for nitrate (e.g., by 1 to 3 permil in the northeastern US) (Garten 1996, Elliott et al. 2009) and ammonium (up to 33 permil) (Heaton et al. 1997).

Line 225- nitrate could also be removed from the soil by biological uptake. This also has potential to be a fractionating process (although evidence for fractionation by directly

by plants under field conditions is limited, mycorrhizal fractionation is likely). You posit later that plant uptake of nitrate is low, but this may not necessarily be the case.

Reply: Thank you. We will add plant and microbial uptake of NO3- in the revision. For example, 'Nitrate can be removed from the ecosystem via denitrification, leaching, and also plant and microbial uptake.' This study area is highly N-limited according to previous N manipulation experiment. Plant would take in both 15N and 14N in N limited areas (Craine et al. 2015). So, the fractionation effect during the plant N uptake could be low.

Line 235- increasing compared to what? This is important.

Reply: The sentence will be modified as 'In addition, a preliminary study using a 15N-labelled incubation experiment also showed that the potential N2 loss rates via denitrification were increasing with precipitation increasing in the semiarid soils'.

Line 244- There are several potential mechanisms for chemodenitrification (see again Homyak 2016)

Reply: Thank you. We will add more discussion on chemodenitrification based on the references provided. Please see our above response to the 'NO losses' (Specific comments 2) discussion).

Line 255- Soper et al (2016, Global Biogeochemical Cycles) did find increased NH3 flux with wetting in an arid system.

Reply: Soper et al. (2016) reported increased NH3 flux 24 hours after a 15 mm artificial rain in soils with pH  7.1, likely due to the stimulation of NH3 production (ammonification) followed by NH3 volatilization. However, the dominant post-rain N loss was still NO loss in that study, likely due to the enhanced nitrification, as this reviewer had emphasized. In our transect within the semi-arid zone spanning a precipitation gradient from 140 mm to 436 mm, and pH decrease from 8.6 to 6.7, plus associated vegetation change and plant NH4+/NO3- uptake, we believe the dominant drivers on soil NH4+

consumption are 1) plant uptake, and 2) nitrification. Nevertheless, the study from Soper et al. (2016) is interesting for our understanding of dryland N loss and will be cited in our revision. Thank you!

Line 256- "First, plant uptake will be enhanced when it is coupled with the microbe-regulating N cycling"- I'm not sure what this means

Reply: Sorry for the confusing. The stimulation of pulse rainfall events to microbes and plant N uptake is different, with a lower stimulation threshold for microbes in extremely dry areas (Dijkstra et al. 2012). Below the MAP threshold, soil microbes may be activated by small rainfall events compared with plants, producing a pulse of high N availability to plants. But if there is an asynchrony in N-cycling via water limitation on plant N uptake, the mineralized N is subject to nitrification and denitrification losses. Above the MAP threshold, these two processes are probably coupled (i.e., microbial mineralized N immediately used by plant), resulting in higher N retention efficiency (Wang et al., 2014). We will rewrite this sentence as well as the entire paragraph to discuss both plant N uptake and nitrification on the consumption of soil $NH_4+$ and its isotopic signal.

Line 268- This doesn't make sense. I don't know of any evidence showing preference for enriched substrates- I would expect it to be exactly the reverse in fact.

Reply: Thank you. To avoid misleading, we will delete this sentence.

Line 324- This paragraph should be rewritten for clarity- right now it's just listing off a bunch of processes and it's confusing. What are the processes that would explain more$NH_4+$, with higher enrichment, at low precipitation? Increasing volatilization with precip explains the concentration gradient, but would induce the opposite isotopic pattern (though it depends really on how much volatilization occurs as a fraction of the standing pool). A greater proportion of atmospheric deposition versus mineralization at low precip might explain the higher 15N. If you invoke fixation by BSCs at low precip, this would also tend to decrease, rather than increase, the 15N at those sites. This also

needs to be clarified- are there BSCs on the transect? However it looks from the gene abundance data like N fix genes increase along the transect. Rethink this paragraph.

Reply: We have rethought the whole paragraph and will rewrite it to address the respective contribution of aerosol deposition and BSC to NH4+ accumulation in the arid zone soils. For example, this paragraph will be modified as, 'There was also a slight NH4+ accumulation in the arid zone soils, with higher 15N enrichment in some sites (Fig. 2b, e). This results might be driven by the mixing of many input and output processes. Input processes mainly include NH4+ deposition (accompanying with nitrate deposition) and ammonification, and output processes include NH3 volatilization and nitrification mentioned above. Ammonium has been shown to be the dominant species in bulk N deposition in China (Liu et al., 2013), thus it could be one of those processes contributing to NH4+ accumulation in the arid zone soils. Dry deposition is the dominant form of deposition in arid climate. Furthermore, higher $\delta$15N values in dry deposition than wet deposition have been reported (Elliott et al., 2009; Garten, 1996; Heaton et al., 1997). In the drylands, biological N fixation is considered to be an important source of N input (Evans and Ehleringer, 1993). With the exception of the biological N fixation by legume plant (Caragana spp.) showed in the same transect (Wang et al., 2014), in this study, we speculated that biological N fixation by BSCs (Wu et al., 2009; Zhuang et al., 2015) also contributed to soil NH4+ pool in the arid zone. We did observe BSCs during sampling in the arid zone soils (personal observation). Besides, the previous research has reported the potential N-fixing activity and potential N input of BSCs in the grasslands of Inner Mongolia (Liu et al., 2009). Biological N fixation by BSCs provided NH4+ with $\delta$15N value around zero, contributing to the pool size of soil NH4+ but not its 15N enrichment.' .

Line 336- Does fractionation during mineralization actually increase with mineralization rate though? I don't recall any evidence for this. Also I think invoking heterotrophic nitrification, when as far as I know there isn't a lot of evidence this is an important process in the field, is a stretch. Maybe remove.

Reply: Though the isotope effect of N mineralization is most often said to be low or negligible, it might be higher than we expected. Our lab recently reported that $\delta$15N values of soil NH4+ were lower than that of bulk soil N by 6-8 permil in two forest soils collected in northern China (Zhang et al. 2015). As had suggested by review 2, they can be as high as 20 permil if one looks at enzymes and their isotope effects that are most likely involved in deamination of organic N forms in cells (Werner and Schmidt 2002). We are not sure about the changes of fractionation effect during mineralization along the precipitation gradient, but fractionation effect could exist. We are not stating that fractionation itself increased with increasing mineralization rate. Clearly nitrification occurs in our study area. Ammonium was 15N depleted relative to bulk soil, indicating that there might be only a small fraction of soil ammonium was oxidized by autotrophic nitrifies. Heterotrophic nitrification therefore could be one of the reasons for the source of soil nitrate, and contributes to the size increase of soil nitrate pool, but not to the 15N signal of soil ammonium. Both the fractionation effect over mineralization and the occurrence of heterotrophic nitrification were the alternative reason for 15N depleted soil ammonium in the semiarid zone soils. Based on those reason, we would like to keep this assumption although we did not test it in this paper, and there were few reports on the fractionation effect of heterotrophic nitrification till now.

Line 359- "Increasing ammonification with increasing MAP both reduced NH3 volatilization" Why would more mineralization reduce volatilization? Unless you mean that volatilization decreases with precip? Again, I'm not sure that this is necessarily true. pH probably the main driver.

Reply: Sorry for the confusing. This sentence will be modified as 'With the increasing of precipitation, both the stimulated ammonification and reduced NH3 volatilization (with low pH) may contribute to 15N depleted soil ammonium pool'.

Line 360- why does more mineralization mean more plant uptake? Plant uptake is likely more a function of water availability. These things likely co-occur, but it's not causal. Perhaps misinterpretation of wording- re-write.

Reply: Sorry for the confusing. In the semiarid zone, with increasing precipitation, plant N uptake was going to couple with N mineralization (Please see response to line 256 above). So N mineralization, nitrification, and plant N uptake all increase with increasing precipitation. We will rewrite this sentence as 'Higher ammonification (N mineralization) would couple with plant N uptake and also favor soil nitrification'.

Figure 8- I think it needs to be clear that the size of arrows indicates qualitative interpretation of these fluxes rather than actual measurement- the presence of pool sizes on the boxes makes this especially necessary. And again, NO loss is likely much more important than anammox and should appear here.

Reply: Thank you for your suggestion. We will add 'qualitative interpretation' in the figure legend. This figure was going to illustrate relative importance of various N processes. The specific N loss was not shown in the figure. We recognized that NO loss was very important in the N losses in these drylands, however, it is part of the nitrification process (as well as part of the denitrification process according to the 'Leaking Pipe Hypothesis'). Similarly, we did not show N2O/N2 loss separately from denitrification. Rather, NO loss would be discussed in specific section (e.g. soil nitrate losses) in the discussion.

Technical corrections: The manuscript contains many examples of awkward or technically incorrect English that can obscure meaning and requires editing by a native English speaker before publication. E.g. in the abstract –'our understanding of' might replace 'understanding about' and 'nitrogen cycling in drylands' rather than 'nitrogen cycling of drylands'. Also 'the patterns and mechanisms of water availability on soil N cycling' doesn't make technical sense. 'Driving' rather than 'driven'. 'Above and below' rather than 'on the two sides of'. 'Preference for' rather than 'preference of', etc. Reply: Thank you very much for your comments and correction. In the revised version of the manuscript, we will make further efforts to improve writing. The reviewer 2 had also give a lot of suggestions on how to improve the writing.

Respectively,

Liu D, Zhu W, and Fang Y, on behalf of all co-authors.

References

Dijkstra, F. A., D. J. Augustine, P. Brewer, and J. C. von Fischer. 2012. Nitrogen cycling and water pulses in semiarid grasslands: are microbial and plant processes temporally asynchronous? Oecologia 170:799-808.

Elliott, E. M., C. Kendall, E. W. Boyer, D. A. Burns, G. G. Lear, H. E. Golden, K. Harlin, A. Bytnerowicz, T. J. Butler, and R. Glatz. 2009. Dual nitrate isotopes in dry deposition: Utility for partitioning NOx source contributions to landscape nitrogen deposition. Journal of Geophysical Research: Biogeosciences 114:n/a-n/a.

Garten, C. T. 1996. Stable nitrogen isotope ratios in wet and dry nitrate deposition collected with an artificial tree. Tellus B 48:60-64.

Heaton, T. H. E., B. Spiro, and S. M. C. Robertson. 1997. Potential canopy influences on the isotopic composition of nitrogen and sulphur in atmospheric deposition. Oecologia 109:600-607.

Lelieveld, J., and F. J. Dentener. 2000. What controls tropospheric ozone? Journal of Geophysical Research: Atmospheres 105:3531-3551.

Wang, C., X. Wang, D. Liu, H. Wu, X. Lü, Y. Fang, W. Cheng, W. Luo, P. Jiang, and J. Shi. 2014. Aridity threshold in controlling ecosystem nitrogen cycling in arid and semi-arid grasslands. Nature communications 5:4799doi:4710.1038/ncomms5799.

Werner, R. A., and H.-L. Schmidt. 2002. The in vivo nitrogen isotope discrimination among organicplant compounds. Phytochemistry 61:465-484.

Zhang, S., Y. Fang, and D. Xi. 2015. Adaptation of micro-diffusion method for the analysis of 15N natural abundance of ammonium in samples with small volume. Rapid Communications in Mass Spectrometry 29:1297-1306.

---

## Author Comment (AC2) · 1 Sep 2016

Dear editor,

It is really our honor to have so many constructive comments and suggestions on our manuscript from the reviewer #2. The kind corrections on writing encourage us to improve our current and future work. In the revised version of this manuscript, we will fix many sentences according to the suggestions, and improve the English writing by asking a native speaker to correct our revised manuscript before resubmission. Please find our specific revisions to each comments below.

Review Biogeosciences Discussion BG-2016-226 The paper "Abiotic versus biotic controls on soil nitrogen cycling in drylands along a 3200 km transect "provides a great dataset on soil N cycling across a precipitation gradient in dryland ecosystems in

[Figure]

China, based on the natural 15N (18O) abundances of bulk soils and ammonium and nitrate, and on the abundances of marker genes involved in N cycling. These novel data allow deep and unprecedented insights into the controls of inorganic N cycling of these ecosystems, and clear trends emerge in abiotic versus biotic controls. The paper therefore addresses relevant questions within the scope of Biogeosciences. Methods and assumptions are valid, and the results definitely sufficient to support the interpretations and implications raised by the authors. The description of Materials and methods and calculations are sufficiently complete. The authors cited relevant work and demonstrate their novel contribution to the field. The title is concise and reflects the content of work, and the abstract concise and complete in summarizing the main points of this study. The presentation/manuscript is well structured and clear, but the language should be edited by a native speaker. The number and quality of references is fair and appropriate, and supplementary material is of high quality and appropriate. Beyond that I have the following comments (according to the lines in the manuscript, the language corrections are by far not complete :)

Reply: Thank you very much for high regard on our work. In the revised version of the manuscript, we will make many efforts to improve the writing.

L36: should read "driving" not driven;

Reply: Will be changed as suggested.

L39: delete significantly;

Reply: Will be changed as suggested.

L41: rewrite "the uptake preference for soil. . .".

Reply: Will be changed as suggested.

L42: soil nitrate loss could also occur by hydrological pathways (leaching) during heavy rain storms.

Reply: Yes, that is true. Leaching is also an alternative pathway by which soil nitrate could be lost. However, given the arid nature of our study sites (annual precipitation from 36 mm to 436 mm) it is less likely the significant pathway of N losses. Therefore, we did not mention it in the abstract section.

L42: rewrite "our study suggests that the shift from abiotic…";

Reply: Will be changed as suggested.

L51: rewrite "factor" not factors;

Reply: Will be changed as suggested.

L54: rewrite "still lack a full understanding of the…"

Reply: Will be changed as suggested.

L61: rewrite "over large scales";

Reply: Will be changed as suggested.

L67: change "are" to "become"

Reply: Will be changed as suggested.

71: change "water-driven" to "hydrological losses by leaching"

Reply: Will be changed as suggested.

73: change to ".. alone is not.."

Reply: Will be changed as suggested.

74: change to "processes that contribute"

Reply: Will be changed as suggested.

77: what is the meaning of "integrate over their characteristics"? Please be more concise.
Reply: This sentence will be modified as "Isotopes in ammonium (NH4+) and nitrate (NO3−) can serve as a proxy record for the N processes in soils because they directly respond to in situ processes and integrate their N cycling characteristics across temporal and spatial scales".

79: rewrite "..provided evidence for: : :"

Reply: Will be changed as suggested.

81: "they cover a different range".

Reply: Will be changed as suggested.

83/84: rewrite "..to study the preferences for plant N uptake"

Reply: Will be changed as suggested.

105: change to "gradient".

Reply: Will be changed as suggested.

109: "gene abundances".

Reply: Will be changed as suggested.

111/112: "with microbially regulated soil processes; and 3) how does soil N cycling: : :".

Reply: Will be changed as suggested.

116: "the climate is: : :".

Reply: Will be changed as suggested.

118: define the aridity index here.

Reply: It will be changed as suggested. 'Aridity index (the ratio of precipitation to potential evapotranspiration) increased from 0.04 to 0.60'.

120: ": : :the three : : :"

Reply: Will be changed as suggested .

124: how do the authors decide which is the peak of soil N transformations? Is that peak vegetation season? Or the short season where the majority of rainfall occurs? Please be more specific here.

Reply: Our soil sampling was conducted from July to August in 2012. The most of the rainfall was occurred at this period of time along the transect. The sentence will be revised as 'Soil sampling was conducted from July to August in 2012, around the peak of plant growing season'.

131: correct"into" to "in", twice.

Reply: Will be changed as suggested.

134: "using a pH meter".

Reply: Will be changed as suggested.

141: "based on the isotopic analysis of nitrous oxide".

Reply: Will be changed as suggested.

142/143: change "into" to "to", three times.

Reply: Will be changed as suggested.

146: rewrite "samples".

Reply: Will be changed as suggested.

148:change "to a Trace: : :".

Reply: Will be changed as suggested.

168: change to "Pearson correlation analysis"

Reply: Will be changed as suggested.

174: it should be "at" not "in" sites.

Reply: Will be changed as suggested.

175: "genes".

Reply: Will be changed as suggested.

177: rewrite "that the soil N status and its controls could be different: : :".

Reply: Will be changed as suggested.

185: "was significantly higher: : :". By the way if I get the numbers correct in the arid zone bulk soil N (soil total N) would be 200 mg N/kg, with nitrate 87 mgN/kg and ammonium 4 mg N/kg, i.e. inorganic N would on average contribute 46% to soil total N, and only 54% on average is bound as organic N in humus?

Reply: Yes, in some sites of the arid zone with extremely limited precipitation, soil N mainly is in inorganic form, and it is mostly driven by inorganic N accumulation by atmospheric deposition (as indicated by the 18O isotopes of soil nitrate in Figure 5a), not by the formation and mineralization of organic matter (and organic N). This is a key point of our result, and also has been observed in the desert soils of northern China (Qin et al. 2012) and northern Chile (Michalski et al. 2004). We will make that clear in our discussion and contrast that to N cycling pattern in the semiarid zone.

188: "supports".

Reply: Will be changed as suggested.

203: "15N depleted relative to their sources".

Reply: Will be changed as suggested.

205: please specify what you mean with "via microbial and plant regulation". 15N depletion of soil ammonium or less 15N enrichment can arise from microbial N mineralization (if this process exerts significant N isotope fractionation) or biological N fixation (causing inputs of N with d15N around 0 to-2 permil). Maybe also atmospheric ammonium/ammonia deposition.

Reply: Thank you. The sentence will be modified as 'The positive values for the 15N enrichment of soil NH4+ support that net NH4+ losses occurred mainly in the arid zone, while the negative values imply that net NH4+ gain (e.g. via microbial mineralization, biological N fixation and/or N deposition) might increase in the semiarid zone, and subsequently reduced the relative 15N enrichment of soil NH4+.' In the later discussion in the manuscript, we will also discuss that higher 15N of deposited ammonium may explain the 15N-enriched soil ammonium in the arid zone. Our preliminary study found that $\delta$15N values of aerosol ammonium in one arid site (Dunhuang in Gansu province, MAP = 46 mm) in northwestern China ranged from 0.35‰ to 36.9‰ with the average of 16.1‰.The similar results have been found in Japan (Kawashima and Kurahashi 2011); $\delta$15N of NH4+ in SPM (suspended particulate matter) ranged from 1.3‰ to 38.5‰ with the average of 11.6‰.These higher $\delta$15N of ammonium in dry deposition may resulted from the exchange of atmospheric ammonia gas and aerosol ammonium (Heaton et al. 1997).

207: rewrite "A positive correlation was: : :"

Reply: Will be changed as suggested.

212: "genes"

Reply: Will be changed as suggested.

213: "rewrite "was measured at all sites"

Reply: Will be changed as suggested.

214:"were found to be. . ."

Reply: Will be changed as suggested.

215 "in the gene abundance of all detected N cycling groups"

Reply: Will be changed as suggested.

217: "dry at the time: : :". "gene abundances in the semiarid zone were: : :"

Reply: Will be changed as suggested.

218: "gene abundances of the five: : :"

Reply: Will be changed as suggested .

219: "potential control of water availability on soil microbial N processes".

Reply: Will be changed as suggested.

223: "water availability drives different patterns" is not meaningful. Please rephrase.

Reply: The sentence will be modified as 'We observed different patterns of N cycling above and below a MAP threshold of 100 mm in this transect'.

223: "at both sides of about MAP = 100 mm" is really not the best phrasing, maybe rather "above and below a MAP threshold of 100 mm".

Reply: Thank you. It will be changed as suggested.

224: "seems to lead to losses of N: : :".

Reply: Will be changed as suggested.

226: "we found direct evidence".

Reply: Will be changed as suggested.

226/227: of course denitrification is a kinetic process. So what? Simply say that denitrification exerts isotope fractionation against the isotopically heavier compounds, ranging between 5 and 25permil: : :"

Reply: Thank you. This sentence will be modified as 'Microbial denitrification exerts

strong isotopic isotopes fractionation against the isotopically heavier compounds, rang-
ing between 5 and 25 ‰ for nitrate nitrogen and oxygen'.

232/233: please specify this sentence on availability of N and O2 supply –to me the
meaning is not clear.

Reply: Thank you. This sentence will be modified as 'Denitrification rate is regulated
by proximal factors that immediately affect denitrifying communities, such as NO3–
concentration and O2 concentration'.

235: "in addition, a preliminary study: : :. an increasingN2 loss via: : :"

Reply: Will be changed as suggested.

240: "in some sites,: : :.. pointing to losses of soil : : :".

Reply: Will be changed as suggested.

241: "after heavy precipitation events".

Reply: Will be changed as suggested.

239-245: the main pattern for soil nitrate at the arid sites is 15N depletion of nitrate
relative to ammonium. Only a few sites had more positive d15N values in nitrate com-
pared to ammonium. The explanation by enhanced denitrification during heavy rain
or chemodenitrification is therefore only secondary. The main pattern has to be ex-
plained – why is soil nitrate 15N depleted relative to ammonium. My best guess is its
production through nitrification which causes ammonium to become 15N enriched and
nitrate 15N depleted (this is also an alternative explanation for the 15N enrichment of
ammonium at many arid sites). I also would not expect large amounts of reduced iron
(FeII) to be present at arid sites. Only in some places denitrification may also play a
role, where nitrate was 15N enriched relative to ammonium. Another input of nitrate
is atmospheric deposition, but its isotopic composition for that region is most probably
unknown (Fig 5(a) indicates that atmospheric nitrate lies between 0 and 5 permil).

Reply: Thank you. We agree with you that 15N depletion for soil nitrate relative to soil ammonium in arid region is in part due to soil nitrification, which exerts a strong isotope fractionation against 15N. Weak denitrification in arid region may have also contributed to low 15N values in soil nitrate. However, we think that main source of soil nitrate in arid region is atmospheric deposition, as indicated by 18O of nitrate in soil nitrate and atmospheric deposition, instead of nitrification, since in those areas, microbial activity may be quite weak even for nitrification. We will make this clear in the revised manuscript.

243: "chemodenitrification is an abiotic process.."

Reply: Will be changed as suggested.

244: change "preserved" to "present"

Reply: Will be changed as suggested.

247: "suggesting losses of: : :"

Reply: Will be changed as suggested .

248: what is the meaning of "ammonia volatilization can be strong for the ammonium loss"???

Reply: Sorry for the confusing. The sentence will be modified as 'Because soil pH was higher in the arid zone (from 7.3 to 9.7; Fig. 6a), NH3 volatilization would play a significant role in NH4+ losses'.

249: "The isotope effect of: : :"

Reply: Will be changed as suggested .

250 "significant negative: : :"

Reply: Will be changed as suggested .

250/251: the alternate explanation is that nitrification can also cause 15N enrichment

of ammonium, and 15N depleted nitrate in many arid soils actually point to a significant role of this process, aside of ammonia volatilization.

Reply: Accepted.

252/253: what does "suggesting the net ammonium gain" mean? Please rephrase.

Reply: The sentence will be fixed as 'in the semiarid zone, soil NH4+ became gradually depleted in 15N relative to the bulk soil N (Fig. 3a), suggesting the contribution of NH4+ input processes, such as soil ammonification in this N limited areas.'

252: soil ammonium "became" gradually 15N depleted relative to: : :

Reply: Done.

252-270: the main pattern of soil ammonium is that it becomes 15N depleted with higher MAP in semiarid sites. This CANNOT be explained with consumption processes such as plant uptake and nitrification, as in both cases (plants and nitrifiers) exert an isotope effect meaning that plants or nitrate become 15N depleted and soil ammonium 15N enriched. An inverse isotope effect has never been shown for any biological process involved in the (production) consumption of ammonium. Lines 268-270 therefore are wrong because microbes will not prefer 15N enriched ammonium during immobilization. The whole paragraph therefore is misleading and has to be rewritten. The explanation can therefore only come from 15N depleted N inputs (biological N fixation, 0 to-2 permil; atmospheric ammonium/ammonia deposition, isotope range for the region unknown?) or its production through mineralization of organic N. Though the isotope effect of N mineralization is most often said to be low or negligible, it might be high if one looks at enzymes and their isotope effects that are most likely involved in deamination of organic N forms in cells (they can be as high as 20 permil). Please consult the respective N isotope reviews such as Werner and Schmidt Phytochemistry 61 (2002)465–484.

Reply: Thank you for your points. First, we agree with you that both the processes of

plant N uptake and nitrification exert isotope effect. However, they may be different in different area. 1) This study area is highly N-limited according to previous N manipulation experiment. Plants would take in both 15N and 14N in N limited areas (Craine et al. 2015). So, the fractionation effect during the plant N uptake could be low. 2) Nitrification includes two types, i.e. autotrophic nitrification and heterotrophic nitrification. To our knowledge, only autotrophic nitrification leaves 15N footprint on the soil ammonium. If the oxidized ammonium by autotrophic nitrification only accounted for a small proportion of total ammonium pool, then this nitrification would not influence 15N of ammonium. Please also see question to line 319-323. Indeed, the isotope effect of N mineralization is most often said to be low or negligible. However, it might be higher than we expected. Our lab recently reported that $\delta$15N values of soil NH4+ were lower than that of bulk soil N by 6-8 permil in two forest soils in northern China (Zhang et al. 2015). As had pointed here, they can be as high as 20 permil if one looks at enzymes and their isotope effects that are most likely involved in deamination of organic N forms in cells (Werner and Schmidt 2002). Thus large 15N depletion in ammonium (by above 10 permil) compared to soil organic matter observed in the semi-arid regions of our study also supports the idea that N mineralization may exert a larger isotope effect as we thought before. We will correct that paragraph regarding this issue.

259: "prefer soil ammonium over nitrate".

Reply: Will be changed as suggested.

263: "demonstrates the ammonium preference of plants".

Reply: Will be changed as suggested.

265: sentence is meaningless – "soil nitrification have been observed to be enhanced with more water widely: : :"?

Reply: Deleted.

271: "we detected anammox genes in these dryland ecosystems".

Reply: Will be changed as suggested.

275: "water-logged".

Reply: Will be changed as suggested.

275/276: "studies of anammox process rates so far failed to: : :".

Reply: Will be changed as suggested.

280: "responsible for gaseous losses: : :".

Reply: Will be changed as suggested.

282:"aeolian".

Reply: Will be changed as suggested.

285: "observed the highest: : :".

Reply: Will be changed as suggested.

287: besides small deposition as dissolved nitrate in rainwater or snow.

Reply: Accepted. It will be incorporated into the main text.

288: "since the d18O: : :".

Reply: Will be changed as suggested.

289: "depends on the d18O: : :".

Reply: Will be changed as suggested.

290: "from the areas closest to: : :".

Reply: Will be changed as suggested.

291: "ranged from: : : to: : :".

Reply: Will be changed as suggested.

294: I don't understand the reasoning behind this sentence, why is atm. O2 and its d18O important. It is not directly expressed in the d18O of NO3- formed in the atmosphere because this is more18O enriched. So...?

Reply: Thank you. The sentence will be fixed as 'In addition, the higher $\delta$18O values of soil NO3– we observed in the arid zone have rarely been reported for nitrified NO3–, according to previous studies'.

285-311: as said before there is also evidence for nitrification in the data set, as in many arid soils nitrate is 15N depleted relative to ammonium, which indicates nitrification also to contribute to soil nitrate accumulation, aside of atmospheric deposition. There are several typos in this paragraph.

Reply: Accepted. The processes of soil nitrification will be incorporated into our revised manuscript. Please also see our response to line 239-245. In addition, we will make efforts to improve writing.

316-318: what does this coincidence of KIE denitrification and d18O of nitrate mean? This is totally dis-connected. Delete.

Reply: Done.

319-323: the gradual 15N depletion of ammonium in itself, but also relative to soil total N indicates that mineralization is the main input process of soil ammonium, and that N mineralization causes 15N fractionation. Obviously nitrification also occurs, but as long as only a small fraction (like 10-20%) of soil ammonium is oxidized by autotrophic nitrifies ammonium would still be 15N depleted relative to bulk soil. Heterotrophic nitrification is another explanation, as stated by the authors.

Reply: We agree with you. See responses above. Thank you!

337: why do the authors believe that soil ammonification was stimulated with higher MAP? Where is the evidence for that? Only the ammonium concentrations?

Reply: Besides of increasing ammonium concentration, we also observed that ammonium 15N was more depleted relative to bulk soil N with higher MAP.

347: was the precipitation range really large?

Reply: The precipitation range was between 36 mm and 436 mm in this study, and may not large enough. This paper focuses on the N cycling in drylands with changing water availability, and especially focus on the available N. From this point of view, the sentence will be modified as 'To the best of our knowledge, our study for the first time showed the pattern of $\delta$15N in soil inorganic N (NH4+ and NO3–) across a precipitation gradient in drylands'.

355: what is phytochemical nitrate loss?

Reply: It is a typo. It should be photochemical nitrate loss.

360: what is "provided lighter N isotope for soil ammonium? And as this sentence states "increasing ammonification reduced ammonia volatilization". How should that happen?

Reply: Sorry for the confusing. This sentence will be modified as 'With the increasing of precipitation, both the stimulated ammonification and reduced NH3 volatilization (with low pH) may contribute to 15N depleted soil ammonium pool'.

Respectively,

Liu D, Zhu W, and Fang Y, on behalf of all co-authors.

References

Craine, J. M., E. Brookshire, M. D. Cramer, N. J. Hasselquist, K. Koba, E. Marin-Spiotta, and L. Wang. 2015. Ecological interpretations of nitrogen isotope ratios of terrestrial plants and soils. Plant and Soil DOI 10.1007/s11104-015-2542-1:1-26.

Heaton, T. H. E., B. Spiro, and S. M. C. Robertson. 1997. Potential canopy influences on the isotopic composition of nitrogen and sulphur in atmospheric deposition. Oecologia 109:600-607.

Kawashima, H., and T. Kurahashi. 2011. Inorganic ion and nitrogen isotopic compositions of atmospheric aerosols at Yurihonjo, Japan: Implications for nitrogen sources. Atmospheric Environment 45:6309-6316.

Michalski, G., J. Böhlke, and M. Thiemens. 2004. Long term atmospheric deposition as the source of nitrate and other salts in the Atacama Desert, Chile: New evidence from mass-independent oxygen isotopic compositions. Geochimica et Cosmochimica Acta 68:4023-4038.

Qin, Y., Y. H. Li, H. M. Bao, F. Liu, K. J. Hou, D. F. Wan, and C. Zhang. 2012. Massive atmospheric nitrate accumulation in a continental interior desert, northwestern China. Geology 40:623-626.

Werner, R. A., and H.-L. Schmidt. 2002. The in vivo nitrogen isotope discrimination among organicplant compounds. Phytochemistry 61:465-484.

Zhang, S., Y. Fang, and D. Xi. 2015. Adaptation of micro-diffusion method for the analysis of 15N natural abundance of ammonium in samples with small volume. Rapid Communications in Mass Spectrometry 29:1297-1306.

---

## Author Response (AR1)

Dear Dr. Michael Weintraub,

Please find enclosed the revision of our manuscript entitled "Abiotic versus biotic controls on soil nitrogen cycling in drylands along a 3200 km transect" (Manuscript # bg-2016-226).

We would like to extend our grateful thanks to reviewers for his/her constructive comments and suggestions to our manuscript. The kind corrections on writing encourage us to improve our current and future work. In the revised version of this manuscript, 1) we rethought our data carefully and fixed many confusing sentences that the reviewer had pointed out; 2) the discussion section has been rewritten according to the reviewers' and your comments. The relevant references have been cited accordingly; and 3) we have also made further efforts to improve the English writing by asking a native speaker to proof our revised manuscript before the resubmission.

Please find our line-by-line response to the review comments below. In addition, we have highlighted the changes in the marked-up manuscript with yellow background.

Thank you again for handling and editing our manuscript!

Respectively,

Yunting Fang, on behalf of all co-authors

CAS Key Laboratory of Forest Ecology and Management

Institute of Applied Ecology, the Chinese Academy of Science, No.72, Wenhua Road, Shenyang, P. R. China, 110016

Phone: +86-24-83970541

Fax: +86-24-83970300

Email: fangyt@iae.ac.cn

**Response to reviewer #1**

General comments

Overall, this manuscript provides an insightful data set that I believe will be of interest to anyone interested in aridland biogeochemistry. The large geographic scale and compound-specific isotopic analysis are especially valuable and the conclusions reached seem valid. Generally, the discussion of the mechanisms driving the observed patterns is thorough. There are some issues that need addressing, primarily in the discussion where several important processes have not been raised (mainly NO production), and there are several mechanisms that do not make sense (perhaps partly as a result of unclear English). This section would benefit from revision. Generally, the figures are clear and informative. Editing for English language is necessary prior to publication.

Reply: Thank you very much for your appreciation of our study. The comments from the reviewer are really helpful. In the revised version of the manuscript, we have rewritten part of the discussion section as suggested by the reviewer. We have also improved general writing by asking a native speaker (Ben Eisenkop from Binghamton University) to carefully edit our revised manuscript before the submission.

Specific comments:

1) Methods: Is it just a coincidence that there is a gap in sampling sites around 100 mm MAP?

Reply: You are correct that it is just a coincidence that a gap of MAP seems appear around 100 mm. Our sampling sites were well-distributed at the distance of about 100 km between two adjacent sites. The threshold happened between site #15 (MAP = 102 mm) and site #16 (MAP = 142 mm). Please see Figure 1 in the main text. The 'unintended gap' serves nicely for us to break the entire gradient into 1) arid zone and 2) semiarid zone for data synthesis and discussion.

2) Discussion: What about loss of NO during nitrification? NO can be the dominant trace gas emitted from arid soils, and would explain loss of ammonium without subsequent appearance of nitrate. For process see Firestone & Davidson (1989) For arid land NO production see Homyak et al (2016, PNAS) and Soper et al (2016, Global Biogeochemical Cycles). This process belongs on Figure 8! A discussion of the isotopic consequences of this process should also be included.

Reply: Thank you for your suggestion. We agree with you that NO emission during N transformation is really important in arid soils. In the revised manuscript, we have added details about this NO losses in the section of 'The losses of nitrate and ammonium'. Please see line 250-252 and line 260-265.

In Figure 8, we tried to compare and contrast the major soil N pools and N transformations (ammonification, nitrification, denitrification, among others) between mainly abiotic driven arid zone soils and biotic dominated semi-arid zone soils. Specific losses of N, including NO, $N_2O$, and $N_2$, are not shown; instead, were discussed in the text.

3)The foliar 15N reflecting 15N of NH4+ in the soil- this could also reflect shifting plant physiology across the significant precipitation gradient, rather than just plant source preference for ammonium. Many aspects of plant internal N cycling likely shift as a function of water availability and would influence foliar 15N. This should at least be acknowledged as an alternative explanation. Also, ammonium shows a larger range of isotopic values along the transect than nitrate, making it less likely that plant 15N would correlate with nitrate 15N anyway.

Reply: Thank you for your suggestion. In the revised manuscript, we have added plant physiology as an alternative reason for the changing pattern of foliar $\delta^{15}N$. Please see line 277-278.

Seen from the data of foliar $^{15}N$ and inorganic $^{15}N$, foliar $^{15}N$ range (-2.1~8.7‰) was basically smaller than that of ammonium (-9~16‰) and nitrate (1.6~13.3‰). Significant $^{15}N$ range for soil $NO_3^-$ did exist but did not correlate to the plant $^{15}N$.

4) Deposition- I think you need to be clear about the difference between wet versus dry deposition (with different isotopic signatures) and how you might expect them to change along the transect. Are there any measures of deposition anywhere on the transect you could mention?

Reply: Good point. In our companion work along this transect (Wang et al. 2014), we reported that rates of bulk N deposition (wet + dry) were increasing from west to east along this transect; data were estimated from a published paper (Lelieveld and Dentener 2000).

We expect that dry deposition could be higher than wet deposition in the arid zone soils, and that in the semiarid zone soils the contribution of wet deposition will increase significantly. In previous studies, higher $\delta^{15}N$ values in dry deposition than in wet deposition had been reported for nitrate (e.g., by 1 to 3 permil in the northeastern US) (Garten 1996, Elliott et al. 2009) and ammonium (up to 33 permil) (Heaton et al. 1997). We have incorporated those points in the revised manuscript, and please see line 331-339.

Line 225- nitrate could also be removed from the soil by biological uptake. This also has potential to be a fractionating process (although evidence for fractionation by directly by plants under field conditions is limited, mycorrhizal fractionation is likely). You posit later that plant uptake of nitrate is low, but this may not necessarily be the case.

Reply: Thank you. We have incorporated uptake of $NO_3^-$ via plant and microbes in the revision. Please see line 222-223.

This study area is highly N-limited according to previous N manipulation experiment. Plant would take in both $^{15}N$ and $^{14}N$ in N limited areas (Craine et al. 2015). So, the fractionation effect during the plant N uptake could be low. Please see our statement in line 278-279.

Line 235- increasing compared to what? This is important.

Reply: The sentence has been modified as 'In addition, our preliminary study of a $^{15}$N-labeled $NO_3^-$ incubation experiment showed that potential $N_2$ losses via denitrification were also increasing with increasing precipitation in the semiarid soils (Liu and Fang, unpublished data)'. Please see line 236-238.

Line 244- There are several potential mechanisms for chemodenitrification (see again Homyak 2016).

Reply: Thank you. We have further discussed chemodenitrification based on the references provided. Please see line 260-265.

Line 255- Soper et al (2016, Global Biogeochemical Cycles) did find increased NH3 flux with wetting in an arid system.

Reply: Soper et al. (2016) reported increased $NH_3$ flux 24 hours after a 15 mm artificial rain in soils with pH ~ 7.1, likely due to the stimulation of $NH_3$ production (ammonification) followed by $NH_3$ volatilization. However, the dominant post-rain N loss was still NO loss in that study, likely due to the enhanced nitrification, as this reviewer had emphasized. In our transect within the semi-arid zone spanning a precipitation gradient from 140 mm to 436 mm, and pH decrease from 8.6 to 6.7, plus associated vegetation change and plant $NH_4^+$/$NO_3^-$ uptake, we believe the dominant drivers on soil $NH_4^+$ consumption are 1) plant uptake, and 2) nitrification. Nevertheless, the study from Soper et al. (2016) is interesting for our understanding of dryland N loss and has been cited in our revision. Thank you! Please see line 270.

Line 256- "First, plant uptake will be enhanced when it is coupled with the microbe-regulating N cycling"- I'm not sure what this means.

Reply: Sorry for the confusing. The stimulation of pulse rainfall events to microbes and plant N uptake is different, with a lower stimulation threshold for microbes in extremely dry areas (Dijkstra et al. 2012). Below the MAP threshold, soil microbes may be activated by small rainfall events compared with plants, producing a pulse of high N availability to plants. But if there is an asynchrony in N-cycling via water limitation on plant N uptake, the mineralized N is subject to nitrification and denitrification losses. Above the MAP threshold, these two processes are probably coupled (i.e., microbial mineralized N immediately used by plant), resulting in higher N retention efficiency (Wang et al., 2014).

We have rewritten the entire paragraph to discuss both plant N uptake and nitrification on the consumption of soil $NH_4^+$ and its isotopic signal. Please see the whole paragraph in line 270-284.

Line 268- This doesn't make sense. I don't know of any evidence showing preference for enriched substrates- I would expect it to be exactly the reverse in fact.

Reply: Deleted.

Line 324- This paragraph should be rewritten for clarity- right now it's just listing off a bunch of processes and it's confusing. What are the processes that would explain moreNH4+, with higher enrichment, at low precipitation? Increasing volatilization with precip explains the concentration gradient, but would induce the opposite isotopic pattern (though it depends really on how much volatilization occurs as a fraction of the standing pool). A greater proportion of atmospheric deposition versus mineralization at low precip might explain the higher 15N.

If you invoke fixation by BSCs at low precip, this would also tend to decrease, rather than increase, the 15N at those sites. This also needs to be clarified- are there BSCs on the transect? However it looks from the gene abundance data like N fix genes increase along the transect. Rethink this paragraph.

Reply: We have rethought the whole paragraph and rewritten it to address the respective contribution of aerosol deposition and BSC to $NH_4^+$ accumulation in the arid zone soils. Please see our revised discussion in line 329-345.

Line 336- Does fractionation during mineralization actually increase with mineralization rate though? I don't recall any evidence for this. Also I think invoking heterotrophic nitrification, when as far as I know there isn't a lot of evidence this is an important process in the field, is a stretch. Maybe remove.

Reply: Though the isotope effect of N mineralization is most often said to be low or negligible, it might be higher than we expected. Our lab recently reported that $\delta^{15}N$ values of soil $NH_4^+$ were lower than that of bulk soil N by 6-8 permil in two forest soils collected in northern China (Zhang et al. 2015). As had also suggested by review #2, the isotope effect of N mineralization can be as high as 20 permil if one looks at enzymes level and their isotope effects that are most likely involved in deamination of organic N forms in cells (Werner and Schmidt 2002). We are not sure about the changes of fractionation effect during mineralization along the precipitation gradient, but fractionation effect could exist. We are not stating that fractionation itself increased with increasing mineralization rate. Please see our modifications in line 348-354.

Clearly nitrification occurs in our study area. Ammonium was $^{15}N$ depleted relative to bulk soil, indicating that there might be only a small fraction of soil ammonium was oxidized by autotrophic nitrifies. Heterotrophic nitrification therefore could be one of the reasons for the source of soil nitrate, and contributes to pool size of soil nitrate, but not to the $^{15}N$ signal of soil ammonium. Both the fractionation effect over mineralization and the occurrence of heterotrophic nitrification were the alternative reason for $^{15}N$ depleted soil ammonium in the semiarid zone soils. Please see our explanation in line 324-328.

Line 359- "Increasing ammonification with increasing MAP both reduced NH3 volatilization" Why would more mineralization reduce volatilization? Unless you mean that volatilization decreases with precip? Again, I'm not sure that this is necessarily true. pH probably the main driver.

Reply: Sorry for the confusing. This sentence has been modified as 'Increasing N mineralization with increasing MAP, accompanied with reduced $NH_3$ volatilization associated with lower pH produce soil $NH_4^+$ pool with lighter N isotopes'. Please see line 372-373.

Line 360- why does more mineralization mean more plant uptake? Plant uptake is likely more a function of water availability. These things likely co-occur, but it's not causal. Perhaps misinterpretation of wording- re-write.

Reply: We have rewritten this sentence as 'Ammonification (N mineralization) supplies $NH_4^+$ for both plant uptake and favour soil nitrification'. Please see line 373-374.

Figure 8- I think it needs to be clear that the size of arrows indicates qualitative interpretation of these fluxes rather than actual measurement- the presence of pool sizes on the boxes makes this especially necessary. And again, NO loss is likely much more important than anammox and should appear here.

Reply: Thank you for your suggestion. We have added 'qualitative interpretation' in the figure legend. This figure was going to illustrate relative importance of various N processes. So, the specific N loss was not shown in the figure. We recognized that NO loss was very important in the N losses in these drylands, however, it is part of the nitrification process (as well as part of the denitrification process according to the 'Leaking Pipe Hypothesis'). Similarly, we did not show $N_2O/N_2$ loss separately from denitrification. Rather, NO loss would be discussed in specific section (e.g. soil nitrate losses) in the discussion (please see line 250-252 and 260-265).

Technical corrections:

The manuscript contains many examples of awkward or technically incorrect English that can obscure meaning and requires editing by a native English speaker before publication. E.g. in the abstract –'our understanding of' might replace 'understanding about' and 'nitrogen cycling in drylands' rather than 'nitrogen cycling of drylands'. Also 'the patterns and mechanisms of water availability on soil N cycling' doesn't make technical sense. 'Driving' rather than 'driven'. 'Above and below' rather than 'on the two sides of'. 'Preference for' rather than 'preference of', etc.

Reply: Thank you very much for your comments and correction. In the revised version of the manuscript, we have made further efforts to improve writing. The reviewer #2 have also given a lot of suggestions on how to improve the writing.

References

Dijkstra, F. A., D. J. Augustine, P. Brewer, and J. C. von Fischer. 2012. Nitrogen cycling and water pulses in semiarid grasslands: are microbial and plant processes temporally asynchronous? Oecologia **170**:799-808.

Elliott, E. M., C. Kendall, E. W. Boyer, D. A. Burns, G. G. Lear, H. E. Golden, K. Harlin, A. Bytnerowicz, T. J. Butler, and R. Glatz. 2009. Dual nitrate isotopes in dry deposition: Utility for partitioning NOx source contributions to landscape nitrogen deposition. Journal of Geophysical Research: Biogeosciences **114**:n/a-n/a.

Garten, C. T. 1996. Stable nitrogen isotope ratios in wet and dry nitrate deposition collected with an artificial tree. Tellus B **48**:60-64.

Heaton, T. H. E., B. Spiro, and S. M. C. Robertson. 1997. Potential canopy influences on the isotopic composition of nitrogen and sulphur in atmospheric deposition. Oecologia **109**:600-607.

Lelieveld, J., and F. J. Dentener. 2000. What controls tropospheric ozone? Journal of Geophysical Research: Atmospheres **105**:3531-3551.

Wang, C., X. Wang, D. Liu, H. Wu, X. Lü, Y. Fang, W. Cheng, W. Luo, P. Jiang, and J. Shi. 2014. Aridity threshold in controlling ecosystem nitrogen cycling in arid and semi-arid grasslands. Nature communications **5**:4799doi:4710.1038/ncomms5799.

Werner, R. A., and H.-L. Schmidt. 2002. The in vivo nitrogen isotope discrimination among organicplant compounds. Phytochemistry **61**:465-484.

Zhang, S., Y. Fang, and D. Xi. 2015. Adaptation of micro-diffusion method for the analysis of $^{15}$N natural abundance of ammonium in samples with small volume. Rapid Communications in Mass Spectrometry**29**:1297-1306.

**Response to reviewer #2**

Review Biogeosciences Discussion BG-2016-226

The paper "Abiotic versus biotic controls on soil nitrogen cycling in drylands along a3200 km transect "provides a great dataset on soil N cycling across a precipitation gradient in dryland ecosystems in China, based on the natural 15N (18O) abundances of bulk soils and ammonium and nitrate, and on the abundances of marker genes involved in N cycling. These novel data allow deep and unprecedented insights into the controls of inorganic N cycling of these ecosystems, and clear trends emerge in abiotic versus biotic controls. The paper therefore addresses relevant questions within the scope of Biogeosciences. Methods and assumptions are valid, and the results definitely sufficient to support the interpretations and implications raised by the authors. The description of Materials and methods and calculations are sufficiently complete. The authors cited relevant work and demonstrate their novel contribution to the field. The title is concise and reflects the content of work, and the abstract concise and complete in summarizing the main points of this study. The presentation/manuscript is well structured and clear, but the language should be edited by a native speaker. The number and quality of references is fair and appropriate, and supplementary material is of high quality and appropriate. Beyond that I have the following comments (according to the lines in the manuscript, the language corrections are by far not complete :)

Reply: Thank you very much for the high regard on our work. In the revised version of the manuscript, we have made many efforts to improve the writing.

L36: should read "driving" not driven;

Reply: Changed as suggested. See line 35.

L39: delete significantly;

Reply: Changed as suggested.

L41: rewrite "the uptake preference for soil…".

Reply: Changed as suggested. See line 40.

L42: soil nitrate loss could also occur by hydrological pathways (leaching) during heavy rain storms.

Reply: Yes, that is true. Leaching is an alternative pathway by which soil nitrate could be lost. However, given the arid nature of our study sites (annual precipitation from 36 mm to 436 mm) it is less likely the significant pathway of N losses. Therefore, we did not mention it in the abstract section.

L42: rewrite "our study suggests that the shift from abiotic…";

Reply: Changed as suggested. See line 41.

L51: rewrite "factor" not factors;

Reply: Changed as suggested. See line 49.

L54: rewrite "still lack a full understanding of the…"

Reply: Changed as suggested. See line 52.

L61: rewrite "over large scales";

Reply: Changed as suggested. See line 59.

L67: change "are" to "become"

Reply: Changed as suggested. See line 65.

L71: change "water-driven" to "hydrological losses by leaching"

Reply: Changed as suggested. See line 68-69.

L73: change to "..alone is not.."

Reply: Changed as suggested. See line 71.

L74: change to "processes that contribute"

Reply: Changed as suggested. See line 72.

L77: what is the meaning of "integrate over their characteristics"? Please be more concise.

Reply: This sentence has been modified as 'Isotopes in ammonium ($NH_4^+$) and nitrate ($NO_3^-$) can serve as a proxy record for the N processes in soils because they directly respond to in situ processes and reflect the processes controlling $NH_4^+$ and $NO_3^-$ production and consumption'. Please see line 73-74.

L79: rewrite "..provided evidence for: : :"

Reply: Changed as suggested. See line 77.

L81: "they cover a different range".

Reply: Changed as suggested. See line 79.

L83/84: rewrite "..to study the preferences for plant N uptake"

Reply: Changed as suggested. See line 81-82.

L105: change to "gradient".

Reply: Changed as suggested. See line 102.

L109: "gene abundances".

Reply: Changed as suggested. See line 106.

L111/112: "with microbially regulated soil processes; and 3) how does soil N cycling: : :".

Reply: Changed as suggested. See line 108-109.

L116: "the climate is: : :".

Reply: Changed as suggested. See line 113.

L118: define the aridity index here.

Reply: It has been modified as 'Aridity index (the ratio of precipitation to potential evapotranspiration) increased from 0.04 to 0.60'. See line 115-116.

L120: ": : :the three : : :"

Reply: Changed as suggested. See line 118.

L124: how do the authors decide which is the peak of soil N transformations? Is that peak vegetation season? Or the short season where the majority of rainfall occurs? Please be more specific here.

Reply: Our soil sampling was conducted from July to August in 2012. The most of the rainfall was occurred at this period of time along the transect. The sentence has been revised as 'Soil sampling was conducted from July to August in 2012, the peak of plant growing season'. Please see line 121.

L131: correct "into" to "in", twice.

Reply: Changed as suggested. See line 129 and 130.

L134: "using a pH meter".

Reply: Changed as suggested. See line 133.

L141: "based on the isotopic analysis of nitrous oxide".

Reply: Changed as suggested. See line 140.

L142/143: change "into" to "to", three times.

Reply: Changed as suggested. See line 141 and 142.

L146: rewrite "samples".

Reply: Changed as suggested. See line 145.

L148:change "to a Trace: : :".

Reply: Changed as suggested. See line 147.

168: change to "Pearson correlation analysis".

Reply: Changed as suggested. See line 167.

174: it should be "at" not "in" sites.

Reply: Changed as suggested. See line 173.

L175: "genes".

Reply: Changed as suggested. See line 174.

L177: rewrite "that the soil N status and its controls could be different: : :".

Reply: Changed as suggested. See line 176.

L185: "was significantly higher: : :". By the way if I get the numbers correct in the arid zone bulksoil N (soil total N) would be 200 mg N/kg, with nitrate 87 mgN/kg and ammonium 4 mg N/kg, i.e. inorganic N would on average contribute 46% to soil total N, and only 54% on average is bound as organic N in humus?

Reply: (1) Please see line 184. (2) Yes, in some sites of the arid zone with extremely limited precipitation, soil N mainly is in inorganic form, and it is mostly driven by inorganic N accumulation by atmospheric deposition (as indicated by the [18]O isotopes of soil nitrate in Figure 5a), not by the formation and mineralization of organic matter (and organic N). This is a key point of our result, and has also been observed in the desert soils of northern China (Qin et al. 2012) and northern Chile (Michalski et al. 2004). Please see our discussion in the whole paragraph form line 297 to 319 and line 331-340.

188: "supports".

Reply: Changed as suggested. See line 187.

203: "15N depleted relative to their sources".

Reply: Changed as suggested. See line 202.

205: please specify what you mean with "via microbial and plant regulation". 15N depletion of soil ammonium or less 15N enrichment can arise from microbial N mineralization (if this process exerts significant N isotope fractionation) or biological N fixation (causing inputs of N with d15N around 0 to-2 permil). Maybe also atmospheric ammonium/ammonia deposition.

Reply: Thank you. The sentence has been modified as 'The positive values for the $^{15}$N enrichment of soil $NH_4^+$ support that net $NH_4^+$ losses occurred mainly in the arid zone, while the negative values imply that net $NH_4^+$ gain (e.g. via microbial mineralization, biological N fixation and/or N deposition) might increase in the semiarid zone, and subsequently reduced the relative $^{15}$N enrichment of soil $NH_4^+$.' Please see line 203-205.

In the later discussion in the manuscript, we also discussed that higher $^{15}$N of deposited ammonium may explain the $^{15}$N-enriched soil ammonium in the arid zone. Our preliminary study found that $\delta^{15}$N values of aerosol ammonium in one arid site (Dunhuang in Gansu province, MAP = 46 mm) in northwestern China ranged from 0.35‰ to 36.9‰, with the average of 16.1‰. The similar results have been found in Japan (Kawashima and Kurahashi 2011); $\delta^{15}$N of $NH_4^+$ in SPM (suspended particulate matter) ranged from 1.3‰ to 38.5‰, with the average of 11.6‰. These higher $\delta^{15}$N of ammonium in dry deposition may resulted from the exchange of atmospheric ammonia gas and aerosol ammonium (Heaton et al. 1997). Please see line 331-339.

207: rewrite "A positive correlation was: : :"
Reply: Changed as suggested. See line 207.

212: "genes"
Reply: Changed as suggested. See line 211.

213: "rewrite "was measured at all sites"
Reply: Changed as suggested. See line 212.

214:"were found to be…"
Reply: Changed as suggested. See line 212.

"in the gene abundance of all detected N cycling groups"
Reply: The sentence has been modified as 'There was a sharp increase (by 8 to 9 fold) in the gene abundance from the arid zone to the semiarid zone'. Please see line 213-214.

217: "dry at the time: : :". "gene abundances in the semiarid zone were: : :"
Reply: Changed as suggested. See line 215.

218: "gene abundances of the five: : :"

Reply: Changed as suggested. See line 216.

219: "potential control of water availability on soil microbial N processes".

Reply: Changed as suggested. See line 217-218.

223: "water availability drives different patterns" is not meaningful. Please rephrase.

Reply: The sentence has been modified as 'We observed different patterns of N cycling above and below a MAP threshold of 100 mm in this 3200 km transect'. Please see line 221.

223: "at both sides of about MAP = 100 mm" is really not the best phrasing, maybe rather "above and below a MAP threshold of 100 mm".

Reply: Thank you. We changed the sentence as suggested. Please see line 221.

224: "seems to lead to losses of N: : :".

Reply: Changed as suggested. See line 222.

226: "we found direct evidence".

Reply: The sentence has been deleted.

226/227: of course denitrification is a kinetic process. So what? Simply say that denitrification exerts isotope fractionation against the isotopically heavier compounds, ranging between 5 and 25permil: : :"

Reply: Thank you. This sentence has be modified as 'Microbial denitrification exerts large fractionation against the isotopically heavier compounds, ranging between 5 and 25‰ for O and N in $NO_3^-$'. Please see line 225-226.

232/233: please specify this sentence on availability of N and O2 supply –to me the meaning is not clear.

Reply: Thank you. This sentence has been modified as 'Denitrification is regulated by proximal factors that immediately affect denitrifying communities, such as $NO_3^-$ concentration and $O_2$ concentration'. Please see line 230-232.

235: "in addition, a preliminary study: : :. an increasingN2 loss via: : :"

Reply: Thank you. For the first suggestion, we would like to keep the expression of 'our preliminary study'. For the second one, we have modified the expression as 'potential $N_2$ losses via denitrification'. Please see line 237.

240: "in some sites,: : :.. pointing to losses of soil : : :".
Reply: Deleted.

241: "after heavy precipitation events".
Reply: Changed as suggested. See line 246.

239-245: the main pattern for soil nitrate at the arid sites is 15N depletion of nitrate relative to ammonium. Only a few sites had more positive d15N values in nitrate compared to ammonium. The explanation by enhanced denitrification during heavy rain or chemodenitrification is therefore only secondary. The main pattern has to be explained – why is soil nitrate 15N depleted relative to ammonium. My best guess is its production through nitrification which causes ammonium to become 15N enriched and nitrate 15N depleted (this is also an alternative explanation for the 15N enrichment of ammonium at many arid sites). I also would not expect large amounts of reduced iron (FeII) to be present at arid sites. Only in some places denitrification may also play a role, where nitrate was 15N enriched relative to ammonium. Another input of nitrate is atmospheric deposition, but its isotopic composition for that region is most probably unknown (Fig 5(a) indicates that atmospheric nitrate lies between 0 and 5 permil).

Reply: Thank you. We agree with you that $^{15}N$ depletion for soil nitrate relative to soil ammonium in arid region is in part due to soil nitrification, which exerts a strong isotope fractionation against $^{15}N$. Weak denitrification in arid region may have also contributed to low $^{15}N$ values in soil nitrate. However, we think that main source of soil nitrate in arid region is atmospheric deposition, as indicated by $^{18}O$ of nitrate in soil nitrate and atmospheric deposition, instead of nitrification, since in those areas, microbial activity may be quite weak even for nitrification. We have further discussed those issues in the whole paragraph, which are from line 242 to 252, and from line 297 to 319.

243: "chemodenitrification is an abiotic process.."
Reply: Changed as suggested. See line 247.

244: change "preserved" to "present"
Reply: Changed as suggested. See line 249.

247: "suggesting losses of: : :"
Reply: Changed as suggested. See line 254.

248: what is the meaning of "ammonia volatilization can be strong for the ammonium loss"???

Reply: Sorry for the confusing. The sentence has been modified as 'we suggest that $NH_3$ volatilization should play a significant role in $NH_4^+$ losses, because soil pH was higher in the arid zone (from 7.3 to 9.7; Fig. 6a)'. Please see line 254-256.

249: "The isotope effect of: : :"

Reply: Changed as suggested. See line 256.

"significant negative: : :"

Reply: Changed as suggested. See line 257.

250/251: the alternate explanation is that nitrification can also cause 15N enrichment of ammonium, and 15N depleted nitrate in many arid soils actually point to a significant role of this process, aside of ammonia volatilization.

Reply: Thank you. We agree with you that $^{15}$N depletion for soil nitrate relative to soil ammonium in arid region is in part due to soil nitrification, which exerts a strong isotope fractionation against $^{15}$N. However, we think that main source of soil nitrate in arid region is atmospheric deposition, as indicated by $^{18}$O of nitrate in soil nitrate and atmospheric deposition, instead of nitrification, since in those areas, microbial activity may be quite weak even for nitrification. We have further discussed those issue in the whole paragraph form line 297 to 319.

252/253: what does "suggesting the net ammonium gain" mean? Please rephrase.

Reply: The sentence has been fixed as 'In the semiarid zone, soil $NH_4^+$ became gradually depleted in $^{15}$N relative to the bulk soil N (Fig. 3a), suggesting the input of $NH_4^+$ (e.g., soil ammonification, N deposition, etc.), while simultaneously $NH_4^+$ was also consumed'. Please see line 266-267.

252: soil ammonium "became" gradually 15N depleted relative to: : :

Reply: The sentence has been modified as '$NH_3$ volatilization should be low due to relatively lower pH compared to those in the arid zone soils'. Please see line 267-268.

252-270: the main pattern of soil ammonium is that it becomes 15N depleted with higher MAP in semiarid sites. This CANNOT be explained with consumption processes such as plant uptake and nitrification, as in both cases (plants and nitrifiers) exert an isotope effect meaning that plants or nitrate become 15N depleted and soil ammonium 15N enriched.

An inverse isotope effect has never been shown for any biological process involved in the (production) consumption of ammonium. Lines 268-270 therefore are wrong because microbes will not prefer 15N enriched ammonium during immobilization. The whole paragraph therefore is misleading and has to be rewritten. The explanation can therefore only come from 15N depleted N inputs (biological N fixation, 0 to-2 permil; atmospheric ammonium/ammonia deposition, isotope range for the region unknown?) or its production through mineralization of organic N. Though the isotope effect of N mineralization is most often said to be low or negligible, it might be high if one looks at enzymes and their isotope effects that are most likely involved in deamination of organic N forms in cells (they can be as high as 20 permil). Please consult the respective N isotope reviews such as Werner and Schmidt Phytochemistry 61 (2002)465–484.

Reply: Thank you for your points. First, we agree with you that both the processes of plant N uptake and nitrification exert isotope effect. However, they may be different in different area. 1) The area in this study is highly N-limited according to previous N manipulation experiment. Plants would take in both $^{15}$N and $^{14}$N in N-limited areas (Craine et al. 2015). So, the fractionation effect during plant N uptake could be low. 2) Nitrification includes two types, i.e. autotrophic nitrification and heterotrophic nitrification. To our knowledge, only autotrophic nitrification leaves $^{15}$N footprint on the soil ammonium. If the oxidized ammonium by autotrophic nitrification only accounted for a small proportion of total ammonium pool, then this nitrification would not influence $^{15}$N of ammonium. We have incorporated those explanations in the manuscript, and please see line 271-284.

Indeed, the isotope effect of N mineralization is most often said to be low or negligible. However, it might be higher than we expected. Our lab recently reported that $\delta^{15}$N values of soil $NH_4^+$ were lower than that of bulk soil N by 6-8 permil in two forest soils in northern China (Zhang et al. 2015). As had pointed here, they can be as high as 20 permil if one looks at enzymes and their isotope effects that are most likely involved in deamination of organic N forms in cells (Werner and Schmidt 2002). Thus large $^{15}$N depletion in ammonium (by above 10 permil) compared to soil organic matter observed in the semi-arid regions of our study also supports the idea that N mineralization may exert a larger isotope effect. We have corrected that paragraph regarding this issue, and please see line 346-357.

259: "prefer soil ammonium over nitrate".
Reply: The expression has changed as 'the dominant plant species might adapt to use soil $NH_4^+$ over $NO_3^-$ as nutrient'. Please see line 273-274.

263: "demonstrates the ammonium preference of plants".
Reply: Deleted.

265: sentence is meaningless – "soil nitrification have been observed to be enhanced with more water widely: : :"?

Reply: Deleted.

271: "we detected anammox genes in these dryland ecosystems".

Reply: The sentence has been modified as 'we detected high anammox gene abundances in these dryland ecosystems'. Please see line 285.

275: "water-logged".

Reply: Changed as suggested. See line 287.

275/276: "studies of anammox process rates so far failed to: : :".

Reply: The sentence has been modified as 'However, the only two anammox studies in drylands so far failed to confirm its importance'. Please see line 288.

280: "responsible for gaseous losses: : :".

Reply: Changed as suggested. See line 292.

282:"aeolian".

Reply: Changed as suggested. See line 294.

285: "observed the highest: : :".

Reply: The sentence has been modified as 'we observed much higher concentrations of soil $NO_3^-$ in the arid zone'. Please see line 297.

287: besides small deposition as dissolved nitrate in rainwater or snow.

Reply: It has been incorporated into the main text. Please see line 299.

288: "since the d18O: : :".

Reply: Please see the correction in line 300.

289: "depends on the d18O: : :".

Reply: Please see the correction in line 302.

290: "from the areas closest to: : :".

Reply: Accepted. See line 302-303.

291: "ranged from: : : to: : :".

Reply: Accepted. See line 303.

294: I don't understand the reasoning behind this sentence, why is atm. O2 and its d18O important. It is not directly expressed in the d18O of NO3- formed in the atmosphere because this is more18O enriched. So...?

Reply: Thank you. The sentence has been fixed as 'The higher $\delta^{18}O$ values of soil $NO_3^-$ we observed in the arid zone have rarely been reported for nitrified $NO_3^-$, according to previous studies'. Please see line 306-307.

285-311: as said before there is also evidence for nitrification in the data set, as in many arid soils nitrate is 15N depleted relative to ammonium, which indicates nitrification also to contribute to soil nitrate accumulation, aside of atmospheric deposition. There are several typos in this paragraph.

Reply: Accepted. The processes of soil nitrification has been incorporated into our revised manuscript. Please also see line 258-262. In addition, we have made great efforts to improve the writing.

316-318: what does this coincidence of KIE denitrification and d18O of nitrate mean? This is totally dis-connected. Delete.

Reply: Done.

319-323: the gradual 15N depletion of ammonium in itself, but also relative to soil total N indicates that mineralization is the main input process of soil ammonium, and that N mineralization causes 15N fractionation. Obviously nitrification also occurs, but as long as only a small fraction (like 10-20%) of soil ammonium is oxidized by autotrophic nitrifies ammonium would still be 15N depleted relative to bulk soil. Heterotrophic nitrification is another explanation, as stated by the authors.

Reply: We agree with you. Please see line 320-328.

337: why do the authors believe that soil ammonification was stimulated with higher MAP? Where is the evidence for that? Only the ammonium concentrations?

Reply: Besides of increasing ammonium concentration, we also observed that ammonium $^{15}N$ was more depleted relative to bulk soil N with higher MAP. Please see further explanation in line 350-354.

347: was the precipitation range really large?

Reply: The precipitation range was between 36 mm and 436 mm in this study, and may not large enough. This paper focuses on the N cycling in drylands with changing water availability, and especially focus on the available N. From this point of view, the sentence has been modified as 'To the best of our knowledge, our study reported for the first time the pattern of $\delta^{15}N$ in soil inorganic N ($NH_4^+$ and $NO_3^-$) across a precipitation gradient from very arid land to semiarid grassland'. Please also see corrections in line 359-360.

355: what is phytochemical nitrate loss?

Reply: Deleted.

360: what is "provided lighter N isotope for soil ammonium? And as this sentence states "increasing ammonification reduced ammonia volatilization". How should that happen?

Reply: Sorry for the confusing. This sentence has been modified as 'Increasing N mineralization with increasing MAP, accompanied with reduced $NH_3$ volatilization associated with lower pH produce soil $NH_4^+$ pool with lighter N isotopes'. Please see line 372-374.

[revised manuscript text omitted]

---

## Author Response (AR2)

Dear Dr. Weintraub:

Please find enclosed the revision of our manuscript entitled "Abiotic versus biotic controls on soil nitrogen cycling in drylands along a 3200 km transect" (Manuscript # bg-2016-226).

We would like to extend our grateful thanks to all reviewers and you for the constructive comments and suggestions to our manuscript. In the revised version of this manuscript, 1) we fixed many confusing sentences that the reviewer had pointed out; 2) the discussion section has been rewritten according to the reviewers'. The relevant references have been cited accordingly; and 3) we have sought professional editorial service (Springer Nature Author Services) for a thorough language editing. All changes are highlighted in the submitted manuscript with yellow background. We hope these will satisfy your request and we are looking forward to the expedited publication of this manuscript on Biogeosciences.

Thank you again for handling and editing our manuscript!

**Reviewer 1**
There are still a number of language errors, and the paper would benefit from an additional close proofread prior to publication, but is otherwise sound
Reply: Thank you for your suggestion. We have found a language company to improve the English writing.

**Reviewer 2**
There's not a lot to say here. It's an excellent study that seems to have benefited from excellent reviews and editorial management. I mainly point out typos below with a few other minor comments. Also I am often critical of figures but these ones are beautifully done. Overall A+ for all involved in the process, especially study authors.
Reply: Thank you very much for your appreciation of our study. Reviewer 2 really gave us a lot of useful and constructive suggestions, from grammar to logic. The kindly provided suggestions on writing encourage us to improve our current and future work. We really appreciated!

L74. typo: controlling = control
Reply: Accepted.

L79 microbially produced
Reply: Done.

L95 Antarctica
Reply: Done.

L 112: 3200 km
Reply: Done.

L 176. recommend change "pointed out" to "suggest" or "indicate." If using suggest, "could be" can be removed.
Reply: Thank you, we changed 'pointed out' to 'indicated'. Please see line 175.

L 177. Rewriting suggestion: Thereafter, we refer to the areas with MAP from 36 mm to 102 mm (15 sites) and from 142 mm to 436 mm (21 sites) as the arid zone and the semiarid zone, respectively.
Reply: Accepted.

L 182. missing a space after NH4+
Reply: Done.

L 285. cool finding. I might mention this in the abstract. I think part of this is that anammox is so energetically inefficient it will occur really slowly and so might not have a huge impact on fluxes.
Reply: Thanks. But we choose not to include this in the abstract as we focus on nitrification and denitrification induced N losses.

L 295. last sentence here doesn't add to paper. make more specific about what should be done or delete.
Reply: We agreed with you and deleted the sentence.

L 312. yes very good analysis, very interesting. I've seen in mountain systems with rich soils that have lots of organic matter the opposite--almost all NO3 of biological origin except for a big flush of atmospheric N from snowmelt. This contrast in a dryland makes a lot of sense.
Reply: Thanks! We hope this paper on dryland N biogeochemistry may stimulate more discussion on the biotic vs. abiotic controls on N cycling across the globe.

L 319. Another point I might make here is that lack of water means that the highly mobile NO3- ions don't leach into streams and groundwater as much as in more mesic areas.
Reply: Thanks! We changed the sentence to 'In the arid zone, extreme dryness and high alkalinity (an average pH of 8.3) might limit microbial activities, as suggested by the low gene abundance involving N transformation (Fig. 4), that combined with the lack of leaching, would facilitate the preservation of $NO_3^-$'. Please see line 316-318.

L 342. Sentence not grammatical. If I understand the point of the sentence, I might rewrite the second phrase to say something like "..., the expected d15N value for NH4+ derived from fixation."
Reply: Thanks! We changed the sentence to 'We found that with decreasing precipitation, the $\delta^{15}N$ of bulk soil N decreased to close to zero, which is the expected $\delta^{15}N$ value for $NH_4^+$ derived from biological N fixation'. Please see line 340-341.

L 344. change "research" to "study"
Reply: Done.

L 367. Effects
Reply: Done.

L 369. Compared
Reply: Done.

L 373. Pools

Reply: Done.

L. 373. "Ammonification (N mineralization) BOTH supplies NH4 + for plant uptake and favourS soil nitrification"
Reply: Accepted.

L 374. enrich THE remaining
Reply: Done.

L 377. I am in favor of deleting the final throwaway line about effects of climate change. A stronger concluding sentence would summarize the key findings of the study, which show that readily interpretable broad-scale patterns of N cycling can be seen across a wide gradient of arid systems.
Reply: After considering the reviewer's recommendation, we choose to keep the global change link but modify the sentence to 'The precipitation regulation of the abiotic vs. biotic controls on N cycling and N losses suggest that global climate changes would have a great impact on these dryland ecosystems.' Please see line 373-375.

**Reviewer 3**
Dear Mike,

I have reviewed the manuscript titled: "Abiotic versus biotic controls on soil nitrogen cycling in drylands along a 3200 km transect." Overall, I think the manuscript provides useful data and information about processes controlling N cycling—it advances understanding of N cycling. I am confident it will be of use to the scientific community exploring how aridity impacts ecosystem N dynamics.
Reply: Thank you very much for your appreciation of our study.

I think it would be useful for a native English speaker to do one more round of editing. I think the data and scope of the paper are great and this begs for the additional editing work to increase overall clarity.
Reply: we have sought professional editorial service (Springer Nature Author Services) for a thorough language editing. All changes are highlighted in the submitted manuscript with yellow background.

I have a few concerns regarding interpretation of results and highlight below issues that I think require attention before publishing.

L62. I suggest reminding the audience what the authors mean by "open" N cycling.
Reply: Thank you! The sentence changed to 'suggesting that N cycling is more open (i.e., more input and output relative to internal cycling) in dryland ecosystems compared with mesic ecosystems'. Please see line 61-62.

L63-65. Good example of a sentence that can use editing
Reply: We have changed the sentence to two separate ones. 'The underlying explanation for openness is when N supply is higher relative to biotic demand, more N is lost through leaching and gaseous N emissions (Austin and Vitousek, 1998). Given that the isotope fractionation during N loss is against the heavier isotope, soils and plant tissues become enriched in $^{15}$N with increasing N losses (Robinson, 2001)'.

L68. I suggest clarifying this sentence as there are drylands in which hydrological N losses are larger than gaseous losses (at least on an annual basis and especially during a wet year).
Reply: We agreed with you and deleted 'instead of hydrological losses'.

L74. "That" can be deleted
Reply: Yes, you are right. To keep a simple sentence, we changed 'controlling' to 'control'.

L130. Perhaps clarify these were frozen. Refrigerator, at least to me, means +4 C.
Reply: It should be +4 C. Please see our corrections in line 129 and 131.

L167. Please specify that SPSS is software.
Reply: Done. Please see line 166.

L175. Figure 4 is out of order because figure 3 has not been mentioned yet.
Reply: Yes, reviewer 2 are technically correct. However, the aim of this paragraph (line 172-177) is to tell the audience why we have the arid and semiarid zone in this study. So, the expression here is just a preface or introduction. Because we would like to keep the current structure in the results section, from soil N concentration,$^{15}$N characteristics to the microbial gene abundance, we changed 'fig. 4' to 'see below'. Please see our corrections in line 174.

L179. This sentence repeats the same information in L173
Reply: The sentence in line 179 was deleted.

L200. It would be useful to edit the sentence so that "positive" is not perhaps interpreted as slope.
Reply: The sentence has been modified as 'The relative $^{15}$N enrichment of soil $NH_4^+$ in the arid zone was mostly above zero, while they were below zero in the semiarid zone (Fig. 3a)'. Please see line 197-199.

We also change Line 204 to 'In a similar way, we found that the relative $^{15}$N-enrichment of $NO_3^-$ were mostly below zero in the arid zone and above zero in the semiarid zone'. Please see line 204-205.

L208. But the 15N-NO3- could have also been enriched through other processes not involving NO3- losses. For example, during the transformation of NH4+ to NO3-, NO2- could have produced NO, enriching the leftover N oxidized to NO3-.
Reply: True. We will have extensive discussion on that later. For now, we simplified that sentence to 'Accordingly, these results suggest that $NO_3^-$ losses may increase when water becomes more available, and progressively enriched residual soil $NO_3^-$ in $^{15}$N'.

L224. But this could have also been produced by NO loss enriching NO2- during nitrification. Can the authors separate between these two processes and be sure of denitrification alone?

Reply: True. Both nitrification and denitrification would promote NO and $N_2O$ loss and also enrich the $NO_2^-$ and $NO_3^-$. However, the dual isotopic analysis data on $\delta^{18}O$ and $\delta^{15}N$ provided direct evidence for denitrification and $NO_3^-$ loss in the semiarid zone.

We cannot rule out the nitrification contribution based on $NO_3^-$ concentration and its $^{15}N$ enrichment. Later, in line 235-237, we addressed this possibility. 'Because gaseous N losses occur during both nitrification (see below) and denitrification, the coupled nitrification and denitrification could maintain low soil $NO_3^-$ concentration while enriching the $^{15}N$ signal'. Further discussion on NO loss is provided in the next two paragraphs.

L232. This is a rough transition because the authors had been discussing denitrification. Nitrification appears suddenly and abruptly.
Reply: Thanks! Rewrite to 'Nitrate can be provided by enhanced microbial processes, including nitrification, when water becomes more available'. Please see line 229-230.

L247-249. To the best of my knowledge, the abiotic reduction of NO3- to NO2- (i.e., the "ferrous wheel hypothesis") has not been confirmed within the context of "natural" soils—if it has, please provide references. Therefore, I suggest minimizing speculation and concentrating on NO2- rather than NO3-; NO2- can react abiotically to produce NO (Heil, J., Vereecken, H., Bruggemann, N., 2016. A review of chemical reactions of nitrification intermediates and their role in nitrogen cycling and nitrogen trace gas formation in soil. European Journal of Soil Science 67, 23-39). Perhaps a mechanism for this NO3- reduction may involve coupled biotic-abiotic processes. I think linking these two papers together might work very well in this manuscript:
Roco, C.A., Bergaust, L.L., Shapleigh, J.P., Yavitt, J.B., 2016. Reduction of nitrate to nitrite by microbes under oxic conditions. Soil Biology & Biochemistry 100, 1-8.
Homyak, P.M., Kamiyama, M., Sickman, J.O., Schimel, J.P., 2016. Acidity and organic matter promote abiotic nitric oxide production in drying soils. Global Change Biology 10.1111/gcb.13507.

By linking these papers it may be possible to explain the increasing δ15N-NO3- in the arid zone as precipitation increases. In extremely arid regions the NO3- signature is dominated by atmospheric deposition but as soils become wetter, the biological signal begins to enrich δ15N-NO3-.

Reply: No, we did not intend to apply the "ferrous wheel hypothesis", which was proposed by Davidson et al. to address the 'rapid' fixation of labeled $NO_3^-$ into soil organic matter (the reduction of $NO_3^-$ by Fe (II) to form $NO_2^-$, then chemically react with soil organic matter).

We have now modified writing to focus more on $NO_2^-$. 'Chemodenitrification is an abiotic process, in which the reduction of $NO_2^-$ to NO and $N_2O$ is coupled to the oxidation of reduced metals (e.g. Fe (II)) and humic substances (Medinets et al., 2015; Zhu-Barker et al., 2015)'. Please see line 244-246.

The Heil et al. review provided several possible coupled biotic-abiotic processes that could affect fates of soil inorganic N and their isotopic signals, which has been cited in this revision, thanks! Please see line 246-248. The Roco et al. paper provided an

important mechanism explaining $NO_3^-$ reduction to $NO_2^-$, which is also cited in the revision. Please see line 249-251.

We have already cited a related Homyak et al. paper. The mechanisms involved in N loss in arid ecosystems can be extremely diverse and complicated and their significances on N budget under field conditions are often poorly known (Heil et al.). Our paper presented a large scale spatial pattern (over a 3200 km transect in China) of inorganic N concentration and related $^{15}NH_4^+$ and $^{15}NO_3^-$ enrichment. The exact mechanisms beyond the patterns are hard to pinpoint and we do not intend to over speculate in the discussion.

L 267. Please clarify: "while simultaneously NH4+ was also consumed." Why would a gradual depletion in 15N-NH4+ suggest consumption? This is opposite to our understanding.
Reply: Deleted.

L 266-284. I have problems understanding how the decrease in 15N-NH4+ as precipitation increases can be explained by plant consumption. I agree that if N is limiting then isotope discrimination would be low (i.e., both 14 and 15 N would be taken up, say at the same rate). However, to cause a decrease in δ15N-NH4+ as precipitation increases you must consume more 15N than 14N or inject more 14N into the system. That mineralization caused fractionation sounds more plausible to me, but this is not well stated and it is confusing. It is critical to the paper to address this well. The shape of figure 3a is controlled by both 15N-NH4+ and bulk soil 15N. I suggest the authors systematically how 15N-NH4+ changes relative to the bulk soil. Doing so would help clarify against figure 2e.
Reply: The focus of discussion here is on $NH_4^+$ consumptions. So, we removed the first sentence to avoid confusing.
We agree with the reviewer's assessment that enhanced N mineralization contributed to the 'dilution' of heavy isotope N. We have talked about this viewpoint in line 344-355.

L300. It would be useful to remind the reader that it is one O from air and two O's from water. The ratio somewhat implies it but it is not clear.
Reply: The sentence has been modified as 'If $NO_3^-$ is formed by nitrification, $NO_3^-$ will contain one O atom from soil $O_2$ and two O atoms from $H_2O$'.

L315-316. Since figure 5a does not show elevated NO3- concentrations, I think it would be useful to clarify that the concentrations are high because these measurements were made in arid regions. Is this right?
Reply: Yes, you are right. The sentence has been modified as 'As shown in Figure 5a, a pronounced trend (green arrow) toward higher $\delta^{18}O$ and lower $\delta^{15}N$ values is obvious for elevated $NO_3^-$ concentrations found in the arid zone soils, which might be the result of mixed $NO_3^-$ from both soil nitrification and atmospheric deposition'. Please see line 313-315.

L325. For the definition of heterotrophic nitrification to work better, it is best to modify this sentence so that the definition is located in the middle of the sentence—as a reminder—rather than as a definition. By starting the sentence with the definition it implies it is a new term not well known by readers of the journal.

Reply: Rewrite as 'Heterotrophic nitrification, a process that oxidizes organic N to $NO_3^-$, bypasses $NH_4^+$'. Please see line 324.

L340-345. The argument about biological soil crusts (BSC) needs to be better developed. Supposedly BSCs contribute to NH4+ inputs as precipitation decreases. However, figure 4 shows the complete opposite: that BSC gene abundance increases in wetter regions and are less important as precipitation decreases. Please elaborate. And this becomes a problem because the authors use the gene abundance data to suggest that nitrification is not likely to dominate N cycling in arid regions. So for nitrification the data are true but somehow for BSCs we are asked to not believe it.

Also, I am confused as to what is meant by "notice that biological N fixation provided NH4+ with the δ15N value around zero." Is the reader supposed to be looking at data showing rates of N fixation in relation to δ15N-NH4+? If so, I don't see those data in the manuscript.
Reply: We believe that reviewer is right here; we should not overplay the significance of BSCs, which we did not directly measured. We have also fixed that confusing sentence.

Changed to: 'In this study, we speculated that biological N fixation by biological soil crusts (BSCs) could contribute to soil $NH_4^+$ pool and soil organic N. We found with decreasing precipitation, the     $\delta^{15}N$ of bulk soil N decreased to close to zero, the expected $\delta^{15}N$ value for $NH_4^+$ derived from biological N fixation. BSCs were observed during soil sampling in the arid zone. A previous research has also reported the potential N-fixing activity and ecological importance of BSCs in soil stability and N availability in the grasslands of Inner Mongolia (Liu et al., 2009)'. Please see line 340-343.

L359. I think the authors have a nice study and great dataset such that they do not need to justify its importance by saying they were first to make this measurement—that is not too useful. What is useful is what we have learned from it.
Reply: Thank you. Agree and changed to 'Our study reported the pattern of $\delta^{15}N$ in soil inorganic N ($NH_4^+$ and $NO_3^-$) across a precipitation gradient from very arid land to semiarid grassland'.

L366. Perhaps say "15N-enriched" instead.
Reply: Accepted.

L600. Figure 1 legend: there is an extra "and"
Reply: We deleted second "and", and started another sentence, 'The dominant plant genera change gradually from shrub ... '.

L609. Figure 3 legend: the first two sentences are identical and can be condensed.
Reply: The first sentence is the figure title, and the second is the explanation for the data calculation. Now, we changed the second one, which started as 'Data in the figures were calculated as…'.

Figure 8. I suggest adding N losses via NO during the transformation of NH4+ to NO3- in both the arid and semiarid zone. The legend of figure 8 requires this information be present. If the authors wish to omit these processes from figure 8, they

must rewrite the legend to be specific as to why they are excluded. The authors are showing several processes removing N from the system (NH3 volatilization, annamox, denitrification) and by that logic NO losses should be identified during the transformation of NH4+ to NO3- . Or at least acknowledge that nitrification can produce losses

Reply: The purpose of Fig. 8 is to contrast abiotic processes (deposition input, volatilization) against biological processes (e.g., nitrification and denitrification), and their different contributions in the arid and semiarid conditions. The loss of NO and $N_2O$ is part of the nitrification and denitrification processes. The losses of N trace gases also often involve coupled biotic-abiotic interactions, as had been extensively discussed in the paper. Consequently, we decide not to include them in the figure but do change figure legend to remind readers the significance of these pathways. Please see our modification in line 636-637.

A suggestion to increase overall clarity: I suggest ending paragraphs with a synthetic "take-home" message summarizing and concluding the paragraph. Ending paragraphs with citations, reinforcing information already known, does not help highlight how this paper is advancing our knowledge.

Reply: We appreciated your useful suggestion! We will go over the entire manuscript, in particular the Discussion section, to see whether such changes are necessary and would improve the overall reading of the paper. Some changes have already been made according to the suggestion of reviewer #2. Thanks!

Respectively,

D. Liu, W. Zhu, and Y. Fang, on behalf of all co-authors.

[revised manuscript text omitted]

640    **Figure 1**

[Figure]

**Figure 2**

[Figure]

**Figure 3**

[Figure]

[Figure]

**Figure 5**

[Figure]

**Figure 6**

[Figure]

660

**Figure 7**

[Figure]

[Figure]

**Figure 8**

665

---

## Author Response (AR3)

Dear Dr. Weintraub:

We made another round of revision based on your advice. The manuscript underwent major revision in Round II, mainly on two parts. The first is language editing. The professional editorial service (Springer Nature Author Services, recommended by the BG) provided excellent improvement to the paper. Their service included editing from two highly-qualified editors, one copy editor and one quality control editor. That was a big help to us non-native speakers. However, to retain final control on the product and be fully responsible to the paper, we took editorial changes one sentence at a time and in some places chose to retain our original words. A few mistakes (although very few) occurred during that revision. In this round, we took time to carefully go over each paragraph of writing, and several of us worked independently on the revision before compiled all changes into the final revised version (R3). Additional changes were also made to improve the clearness of the writing. We submitted both a marked copy shown all changes and a "clean copy". We believe the submitted paper is much improved.

The second part was to mainly addressed comments from Reviewer #3 (although also Reviewer #2, who provided many excellent suggestions). Much of the confusion was in Discussion, in part because $^{15}N$ data of $NH_4^+$ and $NO_3^-$, while important proxy to N transformation and N loss (as well-recognized by all reviewers), cannot provide all the explanation to N biogeochemistry. In Revision 2, we tried to make our argument more logic by following clearly the losses of soil $NO_3^-$ and $NH_4^+$ (4.1) and the sources of soil $NO_3^-$ and $NH_4^+$ (4.2). Multiple pathways contribute to the gaseous loss of N in arid ecosystems, including denitrification (which we had clear isotopic evidence) and nitrification (which have been studied extensively in several recently published papers), among others (e.g., volatilization and plant uptake). However, we do not intend to over-interpret our data, which is based on a large-scale field survey and the measurement of inorganic N concentrations and its isotopic values.

The Firestone and Davidson (1989) paper, when first published, was often referred as the "Leaky pipe model" in the biogeochemistry community. Recent publications, including Homyak et al. PNAS (2016) from the Josh Schimel lab, called it "hole-in-the-pipe" model. They are of course the same thing. To respect the early reference, we change the term back to "leaky pipe model" which likely is more widely known in the field.

We would like to thank you for your patience and if we can do further for this paper or you have any specific suggestions, please do not hesitate to contact us.

Respectively,

D. Liu, W. Zhu, and Y. Fang, on behalf of all co-authors.

[revised manuscript text omitted]